# Consistency Deep Equilibrium Models

Junchao Lin [* 1]   Zenan Ling [* 1]   Jingwen Xu [2]   Robert C. Qiu [1]

## Abstract

Deep Equilibrium Models (DEQs) have emerged as a powerful paradigm in deep learning, offering the ability to model infinite-depth networks with constant memory usage. However, DEQs incur significant inference latency due to the iterative nature of fixed-point solvers. In this work, we introduce the Consistency Deep Equilibrium Model (C-DEQ), a novel framework that leverages consistency distillation to accelerate DEQ inference. We cast the DEQ iterative inference process as evolution along a fixed ODE trajectory toward the equilibrium. Along this trajectory, we train C-DEQs to consistently map intermediate states directly to the fixed point, enabling few-step inference while preserving the performance of the teacher DEQ. At the same time, it facilitates multi-step evaluation to flexibly trade computation for performance gains. Extensive experiments across various domain tasks demonstrate that C-DEQs achieve consistent 2-20× accuracy improvements over implicit DEQs under the same few-step inference budget. Our code is available at https://github.com/landrarwolf/CDEQ.

## 1. Introduction

Deep Equilibrium Models (DEQs) (Bai et al., 2019; 2020), as a class of implicit models, represent a paradigm shift in deep learning by replacing explicit layer-wise depth with an *implicit* infinite depth computation. At their core, DEQs are defined through a *fixed-point* formulation: for a given input $x$, the model directly finds an equilibrium state $z^\star$ satisfying $z^\star = f_\theta(z^\star, x)$, where $f_\theta$ is parameterized by a neural network (NN). Owing to this implicit definition, DEQs exhibit high expressivity and have demonstrated remarkable empirical performance across a broad range of applications,

such as computer vision (Bai et al., 2020; Xie et al., 2022; Hatamizadeh et al., 2023), natural language processing (Bai et al., 2019; Schöne et al., 2025), and graph learning (Gu et al., 2020; Liu et al., 2021; Baker et al., 2023).

Despite their expressive capacity, DEQs incur significant *inference latency* as a result of reliance on iterative solutions of fixed-point equations, where each iteration requires a full evaluation of the DEQ layer. Early efforts (Bai et al., 2019; Winston & Kolter, 2020) mitigate this issue by adopting advanced root-finding methods, such as Broyden's method (Broyden, 1965) and Anderson acceleration (Walker & Ni, 2011). More recent works (Bai et al., 2021b; Lin et al., 2024) further propose hypersolvers with learned initializers and generalized parameterized solvers to accelerate DEQ inference. Nevertheless, even with these advances, existing approaches often require tens of iterations, rendering DEQs substantially slower than explicit NNs with finite depth.

Our goal is to develop DEQ models that enable efficient, few-step inference while retaining the benefits of fixed-point iterations, such as the ability to trade computation for accuracy. We leverage the fundamental connection between fixed point iteration and ODEs to reframe the DEQ iterative solving process as a fixed ODE trajectory toward the equilibrium. Using this trajectory as a teacher, we propose to train a model to consistently map intermediate states directly to the equilibrium. We therefore refer to this model as the Consistency Deep Equilibrium Model (C-DEQ). C-DEQs facilitate inference with only a few network evaluations. Notably, by sequentially applying the consistency mapping over multiple time steps, C-DEQs preserve the advantages of iterative fixed-point solvers, achieving performance gains at the cost of additional computation.

To train C-DEQs, we reframe path-independent DEQs as trajectory-fixed ODEs. Building on this view, finding an equilibrium in a DEQ can be interpreted as discretely sampling a solver-induced fixed-point ODE (FP-ODE). This provides a well-defined teacher trajectory. To facilitate learning along this trajectory, we apply standard consistency parameterization (Song et al., 2023) and, importantly, incorporate the structural prior of the Anderson Acceleration solver (Walker & Ni, 2011) into the student model. This allows the student to refine intermediate estimates rather than rediscover the solver trajectory, accelerating convergence

*Equal contribution [1]School of Electronic Information and Communications, Huazhong University of Science and Technology [2]School of Science, Wuhan University of Technology. Correspondence to: Zenan Ling <lingzenan@hust.edu.cn>.

*Proceedings of the 43$^{rd}$ International Conference on Machine Learning*, Seoul, South Korea. PMLR 306, 2026. Copyright 2026 by the author(s).

and preserving solver-like stability. Finally, we design a distillation loss combining global and local consistency: global consistency anchors predictions to the equilibrium for accurate one-step inference, while local consistency aligns adjacent trajectory points to prevent trajectory drift and support improved multi-step performance.

Our experiments apply C-DEQ to diverse domains with large datasets: WikiText-103 (Merity et al., 2018) for language modeling, ImageNet (Deng et al., 2009) for image classification and the OGB graph benchmarks ogbn-arxiv and ogbn-products (Hu et al., 2020) for graph node classification. Under the same few-step inference budget, C-DEQ achieves consistent 2-20× accuracy improvements over implicit DEQs, while matching or surpassing explicit finite-depth networks and narrowing the latency gap between implicit and explicit models.

Our contributions are summarized as follows.

- We introduce the Consistency Deep Equilibrium Model (C-DEQ), which leverages consistency distillation to map intermediate DEQ states directly to the equilibrium. This enables few-step inference while retaining the benefits of iterative fixed-point solvers.

- We reframe the DEQ iterative process as a fixed solver-induced FP-ODE trajectory to enable well-defined consistency training. Building on this, we incorporate AA's structural prior into the C-DEQ parameterization to accelerate training; and propose local and global consistency losses to stabilize training while balancing single-step and multi-step performance.

- The proposed C-DEQ achieves state-of-the-art acceleration, bringing fixed-point inference into the "one-step" era, with up to 20× faster inference while maintaining the teacher's performance.

## 2. Related Works

**Deep Equilibrium Models.**   DEQs (Bai et al., 2019) represent a significant departure from traditional deep learning architectures by defining the network's output as the fixed point of a nonlinear transformation. This implicit formulation allows for constant-memory training via the implicit function theorem, decoupling the depth of the model from the cost of storing intermediate activations. Many theoretical studies (Ling et al., 2023; Gao et al., 2022; Wu et al., 2024; Sun & Shi, 2024) have investigated the convergence, stability, and expressive capacity of DEQs, highlighting their advantages. Meanwhile, a substantial body of recent works have successfully extended this paradigm to various specialized domains, including sequence modeling (Bai et al., 2019; 2021a), graph-structured data (Gu et al., 2020; Liu et al., 2021; Baker et al., 2023), and high-resolution vi-

sion tasks like optical flow estimation (Bai et al., 2022) and object detection (Wang et al., 2023; Bai et al., 2020). During the training process, several techniques like (Blondel et al., 2022; Geng et al., 2021) have been developed to remarkably improve efficiency of the backward pass. A closely related work (Ding et al., 2023) connects DEQs and Neural ODEs through homotopy continuation, introducing a new implicit model family that interpolates between equilibrium and continuous-time dynamics. Our work takes a different perspective: rather than altering the model formulation, we focus on the solver dynamics of DEQs and accelerate inference by enforcing consistency across truncated iterations.

**Fixed-point Solvers for Deep Equilibrium Models.**   Despite their theoretical elegance and strong empirical performance, reaching the equilibrium state $z^\star$ necessitates an iterative root-finding procedure. Early implementations relied on simple Picard iterations, which often suffer from slow or non-guaranteed convergence, introducing significant inference-time latency and serving as the primary bottleneck for DEQ deployment (Ling et al., 2024; Liu et al., 2024a). To accelerate this process, quasi-Newton methods such as Broyden's method (Broyden, 1965) and Anderson Acceleration (AA) (Walker & Ni, 2011) have been widely adopted as the default forward-pass solvers. Further refinements have introduced monotone operator theory to provide convergence guarantees (Winston & Kolter, 2020) or Jacobian-free approximation techniques to bypass expensive second-order computations (Fung et al., 2022). However, the practical speedups of traditional solvers remain limited (Bai et al., 2021a), maintaining a substantial latency gap between implicit and explicit models.

A parallel line of research explores learning-based acceleration, such as amortized initializations (Huang et al., 2021), which predict the initial state via a lightweight encoder, and Neural DEQ Solvers (Bai et al., 2021a), which replace generic iterations with a learned solver. Despite these advances, such methods still focus on local step-to-step transitions and require multiple sequential iterations to reach equilibrium. Unlike previous works that focus on local iterative refinement, C-DEQ learns a consistency mapping that projects any intermediate state directly to the final equilibrium, effectively leapfrogging the solver trajectory and achieving equilibrium in just a few steps.

**Consistency Models.**   Consistency Models (CMs) (Song et al., 2023), along with subsequent improvements (Song & Dhariwal, 2024; Jain et al., 2025; Heek et al., 2024; Geng et al., 2025; Liu et al., 2024b; Lu & Song, 2024), have emerged as an effective paradigm for accelerating diffusion-based generative models by learning a global projection of the continuous-time probability ODE. Inspired by this principle, we observe that DEQ solvers also define a deter-

ministic ODE trajectory toward equilibrium. Adapting consistency distillation to this fixed-point setting, C-DEQ maps intermediate solver states directly to the final equilibrium, providing a principled way to bypass iterative root-finding and improving efficiency.

A recent closely related work (Kou et al., 2024) applies consistency distillation on a pseudo-Jacobi trajectory for autoregressive generation. While effective, it relies on the naive Jacobi trajectory as the teacher without leveraging the path-independence of the fixed point to define a well-posed target, and employs standard consistency modeling without solver-specific structural priors. C-DEQ, in contrast, distills a solver-informed ODE trajectory and incorporates the structural prior of the advanced AA solver, enabling the student to refine intermediate estimates, preserve solver stability, and achieve few-step inference for general DEQs.

## 3. Methods

In this section, we introduce the framework of Consistency-DEQ (C-DEQ). We begin in Section 3.1 by reinterpreting the iterative root-finding process as a continuous-time flow and addressing the challenge arising from the path independence of DEQs. Building on this foundation, Section 3.3 introduces the C-DEQ parameterization and inference procedure, while Section 3.4 presents the distillation objective and training details.

### 3.1. Reframing Path-Independent DEQs as Trajectory-Fixed ODEs

A DEQ layer (Bai et al., 2019) defines its representation $z^\star$ as the equilibrium state of a non-linear transformation $f_\theta$ parameterized by $\theta$. Given an input $x$, the hidden state $z^\star$ is implicitly defined as the solution to the fixed-point equation:

$$z^\star = f_\theta(z^\star, x), \tag{1}$$

which can be interpreted as the output of an infinite-depth weight-tied network. Instead of infinite computations, the DEQ solves the equilibrium $z^\star$ directly with root-finding:

$$F_\theta(z; x) = f_\theta(z, x) - z = 0.$$

A fundamental challenge in distilling DEQs into a consistency mapping arises from their inherent *path independence* (Anil et al., 2022): while the equilibrium $z^\star$ is uniquely defined, the trajectory used to reach it is not. As illustrated in Figure 1a, multiple trajectories are valid for the same input, making the distillation target ill-defined.

To obtain a deterministic and well-defined distillation target, we observe that the process of finding an equilibrium in a DEQ can be interpreted as discrete sampling of a *trajectory-*

*fixed* Fixed Point ODE (FP-ODE) defined as

$$\frac{dz_t}{dt} = G_\theta(z_t, t), \tag{2}$$

where $G_\theta$ is a conceptual function representing the *solver-induced* dynamics. For example, a simple Picard iteration corresponds to a first-order FP-ODE, while Anderson acceleration (AA) (Anderson, 1965) can be interpreted as inducing history-dependent continuous-time dynamics (Chen et al., 2025). This formulation plays two roles:

- It selects a *deterministic trajectory* for consistency distillation: given a fixed initial condition $z_0$[1], the choice of solver specifies the effective dynamics governing the trajectory toward the equilibrium, whose discrete samples serve as well-defined intermediate targets for consistency distillation.

- It preserves the *same equilibrium* as that of the DEQ in Eq. (1): under well-posedness of the DEQ, the target steady state of the solver-induced FP-ODE in Eq. (2), where $dz_t/dt = 0$, corresponds exactly to the equilibrium state (Ding et al., 2023).

This ODE perspective for DEQs is crucial for enabling consistency distillation (Song et al., 2023) which is inherently defined on continuous-time dynamical systems, where all states along the same trajectory are required to map consistently to the terminal. By reframing DEQ inference as the numerical solution of a solver-induced FP-ODE whose terminal state approaches the equilibrium, we obtain a principled and deterministic path that is particularly well suited for consistency distillation.

### 3.2. Teacher Trajectory

While the FP-ODE in Eq. (2) with a fixed initial condition defines a unique continuous trajectory toward the equilibrium, consistency distillation in practice requires a discrete sequence of intermediate states. We therefore employ a numerical solver to generate a finite teacher trajectory that approximates this flow. In this work, we adopt Anderson acceleration (AA) (Anderson, 1965) for trajectory generation, as it efficiently exploits historical iterates to produce stable and informative intermediate states and often achieves superlinear convergence. We define AA iteration as follows and the complete procedure for AA is detailed in Algorithm 3 of the Appendix C.1.

**Definition 3.1** (Anderson acceleration). Given a fixed-point mapping $f_\theta$ and input $x$, the AA iteration generates the next iterate $z_{k+1}$ as

$$z_{k+1} = \mathcal{S}^{AA}(\{z_{k-m_k+i}\}_{i=0}^{m_k}; f_\theta, x) \tag{3}$$

---

[1]To eliminate variability induced by different initial conditions, we fix $z_0 = 0$.

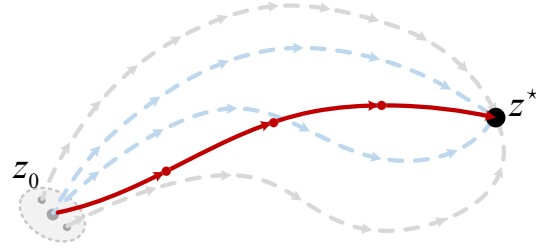

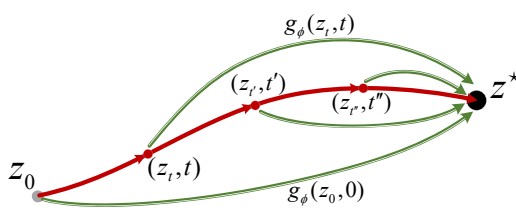

(a) Trajectories of a path-independent DEQ

(b) Consistency mapping of a C-DEQ

*Figure 1.* Illustration of trajectory independence of DEQs and consistency mapping of C-DEQs. (a) Trajectories of a path-independent DEQ. Various initial states (shaded region) and solvers can yield a manifold of a set of trajectories (dashed) toward the fixed point. We propose to fix the initial point and the solver to select *a unique trajectory* (red) converging to $(z_T, T)$. (b) Consistency mapping of a C-DEQ. C-DEQ trains a consistency model $h_\phi$ to map (green) *any* intermediate states along the DEQ trajectory (e.g., $(z_t, t)$ and $(z_{t'}, t')$) directly to the equilibrium state.

where $\mathcal{S}^{\mathrm{AA}}$ is defined by

$$
\begin{aligned}
&\mathcal{S}^{\mathrm{AA}}(\{z_{k-m_k+i}\}_{i=0}^{m_k}; f_\theta, x) \\
&= \beta \sum_{i=0}^{m_k} \alpha_k^i f_\theta(z_{k-m_k+i}) + (1-\beta) \sum_{i=0}^{m_k} \alpha_k^i z_{k-m_k+i}.
\end{aligned}
\tag{4}
$$

where $m_k = \min(k, m)$ denotes the history depth, $m$ is a fixed window size, $\alpha_k$ are least-squares solution over the past few residuals, and $\beta$ is the relaxation parameter controlling the step size.

We denote the teacher trajectory by $\mathcal{T} = \{z_k\}_{k=0}^K$, which is constructed sequentially from $k = 0$ to $K$. Starting from the initialization $z_0$, we generate $z_1, \dots, z_K$ by iteratively applying AA for $k = 1, \dots, K$, where the terminal state $z_K$ serves as an approximation of the equilibrium $z^\star$. To encourage the model to learn the fixed-point property and to "shortcut" the underlying dynamics, we further employ a data augmentation scheme that, with a given probability, replaces intermediate states $z_k$ (excluding the initial and terminal states) with the endpoint $z_K$. Details of the sampling procedure are provided in Appendix C.2.

### 3.3. Consistency DEQs

Below we introduce the parameterization, time mapping and inference procedure of C-DEQs.

We denote C-DEQs by $g_\phi(z_{\leq t}, t, x)$ parameterized by learnable parameters $\phi$. Here, $z_{\leq t}$ denotes the history of solver states up to $t$; in practice, this can be instantiated as the current state $z_t$ alone or a short window of past iterates. As illustrated in Figure 1b, the primary objective is to construct a model $g_\phi$ that consistently maps intermediate states along the same trajectory to the equilibrium.

To ensure the model is anchored at the equilibrium end, we introduce a free-form parameterized network $P_\phi(z_{\leq t}, t, x)$

whose output has the same dimensionality as $z_t$, and define:

$$
g_\phi(z_{\leq t}, t, x) = c_{\mathrm{skip}}(t) z_t + c_{\mathrm{out}}(t) P_\phi(z_{\leq t}, t, x), t \in [\epsilon, T]
\tag{5}
$$

where $c_{\mathrm{skip}}(t)$ and $c_{\mathrm{out}}(t)$ are boundary coefficients, defined as:

$$
c_{\mathrm{skip}}(t) = \left(\frac{t-\epsilon}{T-\epsilon}\right)^\gamma, \quad c_{\mathrm{out}}(t) = 1 - \left(\frac{t-\epsilon}{T-\epsilon}\right)^\gamma,
$$

with $\gamma \geq 1$. As $t \to T$, $c_{\mathrm{skip}}(t) \to 1$ and $c_{\mathrm{out}}(t) \to 0$, ensuring that $g_\phi(z_{\leq t}, t, x) \to z_T \approx z^\star$ near the terminal. As such, the model naturally makes only minor corrections near convergence. Conversely, for early solver states ($t \approx \epsilon$), the prediction is dominated by the neural network $P_\phi$, allowing the model to learn the global mappings required to align initial iterates with the fixed point.

**Time mapping.** The C-DEQ operates on a continuous virtual-time manifold $t \in [\epsilon, T)$, whereas the teacher DEQ solver evolves over discrete iteration steps $k \in \{1, \dots, K\}$. To enable effective distillation, it is therefore necessary to embed the discrete teacher trajectory into a continuous-time domain. We achieve this by introducing a strictly increasing mapping from the solver iteration index $k$ to virtual time $t_k \in [\epsilon, T)$, defined as

$$
t_k = \epsilon + (1 - e^{-\rho k})(T - \epsilon), \quad \rho > 0.
\tag{6}
$$

**AA-Structured Parameterization.** Since our teacher trajectory is discretely generated by Anderson Acceleration (AA), whose updates explicitly depend on past iterates, it naturally injects a solver-specific structural prior into the student, enabling C-DEQs to focus on refining the accelerated estimates rather than relearning the underlying solver trajectory from scratch. Motivated by this, we use $\mathcal{S}^{\mathrm{AA}}$ in Eq. (3) to reparameterize $P_\phi$. Specifically, we construct the input

---

**Algorithm 1** C-DEQ Inference

**Require:** Input $\boldsymbol{x}$, C-DEQ model $g_\phi$, initial state $\boldsymbol{z}_{t_0}$ with initial time $t_0 = \epsilon$, terminal time $T$, inference time schedule $\{t_j\}_{j=1}^{J-1}$ with steps $J \in \mathbb{Z}_{>0}$
1: $P_\phi(\boldsymbol{z}_{t_0}, t_0, \boldsymbol{x}) \leftarrow h_\phi(\boldsymbol{z}_{t_0}, t_0, \boldsymbol{x})$
2: Calculate $\boldsymbol{z}_{t_1} = c_{\text{skip}}(t_0)\boldsymbol{z}_{t_0} + c_{\text{out}}(t_0)P_\phi(\boldsymbol{z}_{t_0}, t_0, \boldsymbol{x})$
3: **if** $J > 1$ **then**
4:     **for** $j = 1$ **to** $J - 1$ **do**
5:         Calculate AA step $P_\phi(\boldsymbol{z}_{\leq t_j}, t_j, \boldsymbol{x}) = \mathcal{S}^{\text{AA}}(\boldsymbol{z}_{t_j}, \boldsymbol{z}_{t_{j-1}}; h_\phi, \boldsymbol{x})$ via Eq. (7)
6:         Calculate $\boldsymbol{z}_{t_{j+1}} = c_{\text{skip}}(t_j)\boldsymbol{z}_{t_j} + c_{\text{out}}(t_j)P_\phi(\boldsymbol{z}_{\leq t_j}, t_j, \boldsymbol{x})$
7:     **end for**
8: **end if**
9: **return** $\boldsymbol{z}^{pred} \leftarrow \boldsymbol{z}_{t_J}$

---

**Algorithm 2** Consistency Distillation Training for C-DEQ

**Require:** Cached trajectory dataset $\mathcal{T}$, C-DEQ model $g_\phi(\cdot; \cdot, \boldsymbol{x})$ with initial parameter $\phi$ and input $\boldsymbol{x}$, metric $d(\cdot, \cdot)$, coefficients $\lambda_1$ and $\lambda_2$, learning rate $\eta$, and EMA rate $\mu$
1: $\phi^- \leftarrow \phi$
2: **repeat**
3:     Sample $\{\boldsymbol{z}_{t_k}\}_{k=0}^K \sim \mathcal{T}$ and $k \sim \mathcal{U}\{1, \dots, K\}$
4:     $\mathcal{L}_{\text{global}}(\phi; \boldsymbol{\theta}) \leftarrow \mathbb{E}_{t_k}\big[d\big(g_\phi(\boldsymbol{z}_{t_k}, t_k, \boldsymbol{x}), \boldsymbol{z}_K\big)\big]$
5:     $\mathcal{L}_{\text{local}}(\phi, \phi^-; \boldsymbol{\theta}) \leftarrow$
        $\mathbb{E}_{t_k, t_{k-1}}\big[d\big(g_\phi(\boldsymbol{z}_{t_k}, t_k, \boldsymbol{x}), g_{\phi^-}(\boldsymbol{z}_{t_{k-1}}, t_{k-1}, \boldsymbol{x})\big)\big]$
6:     $\mathcal{L}_{\text{distill}} \leftarrow \lambda_1 \mathcal{L}_{\text{global}} + (1 - \lambda_1)\mathcal{L}_{\text{local}} + \lambda_2 \mathcal{L}_{\text{task}}$
7:     $\phi \leftarrow \phi - \eta \nabla_\phi \mathcal{L}_{\text{distill}}(\phi, \phi^-; \boldsymbol{\theta})$
8:     $\phi^- \leftarrow \text{stopgrad}(\mu\phi^- + (1 - \mu)\phi)$
9: **until** convergence

---

using a two-step solver history, i.e., $\boldsymbol{z}_{\leq t_k} = \{\boldsymbol{z}_{t_k}, \boldsymbol{z}_{t_{k-1}}\}$, in which case, $\boldsymbol{\alpha}_k = [\alpha_{t_k}, 1 - \alpha_{t_k}]$. We then reparameterize $P_\phi(\boldsymbol{z}_{\leq t}, t, \boldsymbol{x})$ in Eq. (5) as:

$$P_\phi(\boldsymbol{z}_{\leq t}, t, \boldsymbol{x}) = \mathcal{S}^{\text{AA}}(\boldsymbol{z}_{t_k}, \boldsymbol{z}_{t_{k-1}}; h_\phi, \boldsymbol{x}) \tag{7}$$

where $h_\phi(\boldsymbol{z}_t, t, \boldsymbol{x})$ is a learnable function that maps $\boldsymbol{z}_t$ at time $t$ to an estimate compatible with the AA update.

In our implementation, we further parameterize $h_\phi$ using a backbone network consistent with $f_\theta$, augmented with a time-dependent readout head $\boldsymbol{t}$. This head is obtained by broadcasting the scalar time $t$ to match the spatial resolution of $\boldsymbol{z}_t$. Specifically, the operation is defined as:

$$h_\phi(\boldsymbol{z}_t, t, \boldsymbol{x}) = \boldsymbol{W}[f_{\phi'}(\boldsymbol{z}_t, \boldsymbol{x}) \| \boldsymbol{t}], \tag{8}$$

where $\phi$ is partitioned into learnable parameters $\boldsymbol{W}$ and $\phi'$. Here, $f_{\phi'}$ shares the same network architecture as $f_\theta$ but with different parameters, $[\cdot \| \cdot]$ denotes concatenation along the feature dimension, and $\boldsymbol{W}$ projects the concatenated input back to the latent dimensionality.

**C-DEQ Inference.** With a trained C-DEQ $h_\phi$, given an input $\boldsymbol{x}$ and an initial state $\boldsymbol{z}_0$ (as defined in Section 3.1), the equilibrium prediction is obtained directly via Eqs. (5) and (7). This requires only one forward pass through C-DEQ and therefore predicts the equilibrium state in a single step. In particular, C-DEQ facilitates multi-step inference by chaining the consistency mapping according to the time schedule $\{t_j\}_{j=1}^{J-1}$ with $J$ steps. As illustrated in Algorithm 1, each iteration of the inference loop unfolds in two phases: first, we derive the refined intermediate state via AA step using the current state at time $t_j$; subsequently, we predict the next consistency state $\boldsymbol{z}_{j+1}$ via Eq. (5). In practice, we select a monotonically increasing time schedule here, details are provided in Appendix C.3.

## 3.4. Distillation Training

The training objective of C-DEQ consists of two complementary terms designed to ensure that the student model $h_\phi$ is both stable and accurate. Specifically, given an input $\boldsymbol{x}$ and a sampled time index $k \sim \mathcal{U}\{1, \dots, K\}$ from the trajectory $\mathcal{T}$, we define the distillation loss as a combination of local and global constraints.

**Consistency Loss.** To ensure that C-DEQ's predictions at each time step map to the equilibrium, we introduce the *global consistency* loss:

$$\mathcal{L}_{\text{global}}(\phi; \boldsymbol{\theta}) := \mathbb{E}_{t_k}\big[d\big(g_\phi(\boldsymbol{z}_{t_k}, t_k, \boldsymbol{x}), \boldsymbol{z}_K\big)\big], \tag{9}$$

where $d(\cdot, \cdot)$ is a distance metric (e.g., MSE) and $\boldsymbol{z}_K$ is the equilibrium produced by the teacher solver. This loss serves as a global anchor, providing a direct distillation signal that maps intermediate states to the equilibrium.

While global consistency enables accurate one-step approximation of the equilibrium, it does not, by itself, guarantee stability under repeated application of the learned model. To address this limitation, we introduce a *local consistency* loss, which enforces model predictions to be invariant across adjacent points along the solver's trajectory, defined as

$$\begin{aligned}\mathcal{L}_{\text{local}}(\phi, \phi^-; \boldsymbol{\theta}) := \\ \mathbb{E}_{t_k, t_{k-1}}\big[d\big(g_\phi(\boldsymbol{z}_{t_k}, t_k, \boldsymbol{x}), g_{\phi^-}(\boldsymbol{z}_{t_{k-1}}, t_{k-1}, \boldsymbol{x})\big)\big],\end{aligned} \tag{10}$$

where $t_k$ and $t_{k-1}$ are successive time steps in the discrete schedule $\{t_k\}_{k=1}^K$, and $\phi^-$ denotes a running average of the past values of $\phi$ during optimization. In practice, we update the model's parameters $\phi$ by stochastic gradient descent, while updating $\phi^-$ with exponential moving average (EMA) (Klinker, 2011) as

$$\phi^- \leftarrow \text{stopgrad}(\mu\phi^- + (1 - \mu)\phi), \ \mu \in [0, 1).$$

By aligning the estimates from a state $z_{t_k}$ with its immediate predecessor $z_{t_{k-1}}$, we encourage the consistency function to learn a smooth flow. This design is crucial for multi-step refinement during inference, as it ensures consecutive steps close and stay on a stable and smooth trajectory.

We distinguish two forms of consistency. Local consistency enforces alignment between consecutive steps, promoting stability under repeated application of the learned model, while global consistency anchors each state to the equilibrium. Empirically, enforcing only local consistency preserves multi-step stability but leads to inaccurate one-step inference due to degenerate solutions, whereas enforcing only global consistency yields accurate one-step predictions at the expense of multi-step consistency. Combining both objectives is therefore necessary to achieve accurate and stable few-step inference. Interestingly, this observation is consistent with previous findings (Bai et al., 2025) in CMs.

Further, we augment the consistency objective with a lightweight task-level regularizer $\mathcal{L}_{\text{task}}$, derived from downstream ground-truth labels, which encourages intermediate states to preserve task-relevant information and stabilizes training (see Appendix C.4). Therefore, we combine these objectives and the total loss for training a C-DEQ is

$$\mathcal{L}_{\text{distill}} = \lambda_1 \mathcal{L}_{\text{global}} + (1 - \lambda_1)\mathcal{L}_{\text{local}} + \lambda_2 \mathcal{L}_{\text{task}}, \quad (11)$$

where $\lambda_1 \in [0, 1]$ balances equilibrium anchoring and solver-time consistency and $\lambda_2$ controls task-level regularizer. In practice, we find that combining the distillation objectives yields better generalization than either term alone.

## 4. Experiments

**Benchmarks and Setup.** We evaluate C-DEQ in terms of performance, efficiency, and scalability across three modalities on a suite of large-scale, widely adopted benchmarks. For natural language processing, we use WikiText-103 (Merity et al., 2016), a standard testbed for long-range sequence modeling. For computer vision, we benchmark on ImageNet (Deng et al., 2009). For graph learning, we consider the ogbn-arxiv citation network and the ogbn-products co-purchasing graph (Hu et al., 2020). We report test perplexity on WikiText-103, node classification accuracy on the OGB benchmarks, and top-1/top-5 accuracy on ImageNet. All training and inference are conducted on servers equipped with an Intel(R) Xeon(R) Gold 6138 CPU @ 2.00 GHz and 8 NVIDIA RTX 3090 GPUs. Further details on datasets, baselines, and implementation are provided in Appendix A.

**Baselines.** We compare C-DEQ against a broad set of competitive baselines, spanning both traditional explicit architectures and DEQ-style implicit models. For natural language processing, we include TCN (Bai, 2018), Gated ConvNet (GCNN) (Dauphin et al., 2017), AWD-QRNN (Merity et al.,

2018), and Transformer-XL (Dai et al., 2019), as well as DEQ-based Transformer variants, including DEQ (Bai et al., 2019) and HyperDEQ (Bai et al., 2021a). For large-scale graph learning, we benchmark against IGNN (Gu et al., 2020), EIGNN (Liu et al., 2021), MIGNN (Baker et al., 2023), and IGNN-Solver (Lin et al., 2024). For large-scale image classification, we compare with AlexNet (Krizhevsky et al., 2012), Inception-V2 (Ioffe & Szegedy, 2015), Single-stream DEQ (Bai et al., 2019), and MDEQ (Bai et al., 2020). Our evaluation primarily focuses on implicit models, as our C-DEQ is explicitly designed to accelerate implicit inference while retaining strong explicit baselines as reference.

### 4.1. Large-scale Experiments on Various Tasks

To evaluate the effectiveness of C-DEQ, we report wall-clock inference time under identical experimental settings (e.g., the same input scale), rather than the number of function evaluations (NFEs) used in prior work (Chen et al., 2018; Song et al., 2023). For baselines, we use the official pretrained weights and evaluation code without retraining or special tuning for small-NFE settings. The low NFE performance of existing DEQ variants reflects their known dependence on iterative equilibrium solving. See Appendix A for additional implementation details and efficiency analyses, including memory usage, trajectory generation time cost and caching overhead, and additional distillation cost.

**Experiments on Language Modeling.** In Table 1, we report the sequence modeling performance and inference efficiency of C-DEQ and competing baselines on the large-scale WikiText-103 dataset under varying numbers of NFEs. These results demonstrate that C-DEQ can rapidly achieve high-quality outputs within only a few inference iterations, making C-DEQ computationally viable for large-scale, real-time natural language processing tasks. (1) C-DEQ achieves substantial perplexity improvements without requiring a large number of function evaluations. In particular with one step, C-DEQ attains a perplexity that is within an absolute gap of only 2.7 to the explicit TCN model. Compared to the explicit state-of-the-art Transformer with 8 steps, there is only an absolute difference of 2.1 in perplexity. (2) C-DEQ provides a dramatic leap in inference performance and cost. Specifically, compared to DEQ and HyperDEQ at one step, the inference time of C-DEQ significantly outperforms other DEQs up to $14.6\times$, while performance achieving up to $5.3\times$ improvement. (3) Increasing NFE yields *consistent* perplexity reductions. In particular, with only 8 NFEs, C-DEQ achieves a perplexity of $26.43$ in $0.37$s, whereas DEQ (Bai et al., 2019) requires $2.27$s to reach a comparable performance level, amounting to a $6.1\times$ speedup. This result suggests that our multi-step sampling strategy effectively rectifies the solver trajectory and substantially compresses the iterative process for language modeling.

*Table 1.* **Word-level Language Modeling on WikiText-103.** We compare C-DEQ against traditional and learned solvers across various NFEs. Performance is measured by test perplexity (lower is better) and efficiency is measured by inference time(s) per batch.

| Model | # Params | NFE | Test PPL ($\downarrow$) | Infe. (s) |
|---|---|---|---|---|
| Generic TCN (Bai, 2018) | 150M | - | 45.2 | 0.74 |
| GCNN (Dauphin et al., 2017) | 230M | - | 37.2 | 0.30 |
| AWD-QRNN (Merity et al., 2018) | 159M | - | 33.0 | 0.11 |
| Transformer-XL (Dai et al., 2019) | 165M | - | 24.3 | 0.13 |
| DEQ (Bai et al., 2019) | 168M | 1 | 255.94 | 0.09 |
| HyperDEQ (Bai et al., 2021a) | 168M | 1 | 70.19 | 0.73 |
| **C-DEQ (Ours)** | 170M | 1 | **47.90** | **0.05** |
| DEQ (Bai et al., 2019) | 168M | 2 | 223.40 | 0.17 |
| HyperDEQ (Bai et al., 2021a) | 168M | 2 | 51.31 | 0.80 |
| **C-DEQ (Ours)** | 170M | 2 | **38.98** | **0.09** |
| DEQ (Bai et al., 2019) | 168M | 8 | 104.26 | 0.65 |
| HyperDEQ (Bai et al., 2021a) | 168M | 8 | 31.37 | 1.21 |
| **C-DEQ (Ours)** | 170M | 8 | **26.43** | **0.37** |

*Table 2.* **Image Classification on ImageNet.** We compare C-DEQ against traditional and learned solvers across various NFEs. Performance is measured by test accuracy (higher is better) and efficiency is measured by inference time(s) per batch.

| Model | # Params | NFE | Top1 Acc. | Top5 Acc. | Infe. (s) |
|---|---|---|---|---|---|
| AlexNet (Krizhevsky et al., 2012) | 238M | - | 57.0 | 80.3 | 0.75 |
| Inception-V2 (Ioffe & Szegedy, 2015) | 12M | - | 74.8 | 92.2 | 1.37 |
| HRNet-W18-C (Wang et al., 2020) | 21M | - | 76.8 | 93.4 | 1.64 |
| DEQ (Bai et al., 2019) | 18M | 1 | 1.17 | 4.64 | **0.48** |
| HyperDEQ (Bai et al., 2021a) | 18M | 1 | 5.46 | 18.11 | 1.09 |
| MDEQ (Bai et al., 2020) | 18M | 1 | 1.44 | 4.24 | 0.57 |
| **C-DEQ (Ours)** | 18M | 1 | **47.12** | **68.71** | 0.52 |
| DEQ (Bai et al., 2019) | 18M | 2 | 8.12 | 19.43 | **0.67** |
| HyperDEQ (Bai et al., 2021a) | 18M | 2 | 27.82 | 41.59 | 1.34 |
| MDEQ (Bai et al., 2020) | 18M | 2 | 9.70 | 23.31 | 0.70 |
| **C-DEQ (Ours)** | 18M | 2 | **58.28** | **77.10** | 0.69 |
| DEQ (Bai et al., 2019) | 18M | 8 | 64.13 | 82.49 | **0.85** |
| HyperDEQ (Bai et al., 2021a) | 18M | 8 | 69.90 | 84.12 | 2.90 |
| MDEQ (Bai et al., 2020) | 18M | 8 | 71.29 | 89.57 | 0.98 |
| **C-DEQ (Ours)** | 18M | 8 | **74.04** | **91.45** | 0.87 |

*Table 3.* **Large-scale Graph Node Classification on Ogbn-Arxiv and Ogbn-products.** We compare C-DEQ against explicit and implicit GNNs across various NFEs. Performance is measured by Acc. and efficiency is measured by inference time(s) per batch.

| Model | # Params | NFE | Test Acc. ($\uparrow$) | Infe. (s) |
|---|---|---|---|---|
| GCN (Kipf & Welling, 2017) | 1.46M | - | 71.56 | 0.05 |
| GAT (Veličković et al., 2018) | 1.44M | - | 71.10 | 0.06 |
| GCNII (Chen et al., 2020) | 2.15M | - | 72.74 | 0.08 |
| GPM (Wang et al., 2025) | 1.40M | - | 72.89 | 0.78 |
| IGNN (Gu et al., 2020) | 0.15M | 1 | 8.61 | **0.03** |
| EIGNN (Liu et al., 2021) | 0.15M | 1 | 11.30 | 0.05 |
| MIGNN (Baker et al., 2023) | 0.16M | 1 | 8.23 | 0.06 |
| IGNN-Solver (Lin et al., 2024) | 0.16M | 1 | 35.32 | 0.13 |
| **C-DEQ (Ours)** | 0.16M | 1 | **56.81** | 0.05 |
| IGNN (Gu et al., 2020) | 0.15M | 2 | 13.80 | **0.05** |
| EIGNN (Liu et al., 2021) | 0.15M | 2 | 16.37 | 0.07 |
| MIGNN (Baker et al., 2023) | 0.16M | 2 | 15.04 | 0.09 |
| IGNN-Solver (Lin et al., 2024) | 0.16M | 2 | 52.32 | 0.15 |
| **C-DEQ (Ours)** | 0.16M | 2 | **67.48** | 0.08 |
| IGNN (Gu et al., 2020) | 0.15M | 5 | 45.93 | 0.18 |
| EIGNN (Liu et al., 2021) | 0.15M | 5 | 47.25 | **0.13** |
| MIGNN (Baker et al., 2023) | 0.16M | 5 | 43.30 | 0.15 |
| IGNN-Solver (Lin et al., 2024) | 0.16M | 5 | 65.26 | 0.19 |
| **C-DEQ (Ours)** | 0.16M | 5 | **71.40** | 0.16 |

| Model | # Params | NFE | Test ACC ($\uparrow$) | Infe. (s) |
|---|---|---|---|---|
| GCN (Kipf & Welling, 2017) | 2.55M | - | 75.64 | 1.30 |
| GAT (Veličković et al., 2018) | 2.62M | - | 79.45 | 1.46 |
| GCNII (Chen et al., 2020) | 3.79M | - | 80.20 | 1.43 |
| GPM (Wang et al., 2025) | 2.45M | - | 82.62 | 5.54 |
| IGNN (Gu et al., 2020) | 0.17M | 1 | 9.12 | **0.73** |
| EIGNN (Liu et al., 2021) | 0.17M | 1 | 11.96 | 0.95 |
| MIGNN (Baker et al., 2023) | 0.18M | 1 | 8.71 | 0.98 |
| IGNN-Solver (Lin et al., 2024) | 0.18M | 1 | 37.41 | 2.19 |
| **C-DEQ (Ours)** | 0.18M | 1 | **60.15** | 1.39 |
| IGNN (Gu et al., 2020) | 0.17M | 2 | 11.25 | 1.42 |
| EIGNN (Liu et al., 2021) | 0.17M | 2 | 14.65 | **1.02** |
| MIGNN (Baker et al., 2023) | 0.18M | 2 | 11.53 | 1.19 |
| IGNN-Solver (Lin et al., 2024) | 0.18M | 2 | 54.04 | 4.23 |
| **C-DEQ (Ours)** | 0.18M | 2 | **69.68** | 1.86 |
| IGNN (Gu et al., 2020) | 0.17M | 10 | 38.87 | 6.56 |
| EIGNN (Liu et al., 2021) | 0.17M | 10 | 40.29 | **1.58** |
| MIGNN (Baker et al., 2023) | 0.18M | 10 | 36.89 | 2.87 |
| IGNN-Solver (Lin et al., 2024) | 0.18M | 10 | 68.47 | 9.30 |
| **C-DEQ (Ours)** | 0.18M | 10 | **76.55** | 3.85 |

**Experiments on Image Classification.** In Table 2, we report the ImageNet classification accuracy and efficiency of C-DEQ and competing implicit baselines under varying inference budgets (NFEs). The results suggest that: (1) C-DEQ attains surprisingly strong accuracy even with very few function evaluations: with only NFE = 1, it already achieves $47.12\%$ Top-1 and $68.71\%$ Top-5 accuracy; with NFE = 8, it further reaches $74.04\%$ Top-1 and $91.45\%$ Top-5, approaching explicit SOTA performance (e.g., within an absolute gap of 0.76 Top-1 / 0.75 Top-5 to Inception-V2) under a comparable parameter budget. (2) Under the same parameter regime (18M), C-DEQ consistently outperforms DEQ variants at matched NFEs while remaining efficient. In particular, at NFE = 8, C-DEQ improves Top-1 accuracy by $+9.91$, $+4.14$, and $+2.75$ points over DEQ, HyperDEQ, and MDEQ, respectively, and is $\sim 3.3\times$ faster than Hyper-DEQ (0.87s vs. 2.90s per batch) while being comparable to DEQ/MDEQ in runtime. (3) Increasing NFE yields consistent gains (Top-1: $47.12 \to 58.28 \to 74.04$; Top-5: $68.71 \to 77.10 \to 91.45$ for NFE $= 1, 2, 8$), validating that relative to iterative root-finding, our multi-step inference

recovers high-quality equilibrium solutions at lower cost on image classification tasks.

**Experiments on Graph Node Classification.** Table 3 summarizes the trade-off of accuracy and efficiency on ogbn-arxiv and ogbn-products. C-DEQ consistently outperforms prior implicit baselines at the same inference budget: on ogbn-arxiv, it improves from $56.81\%$ (NFE = 1) to $71.40\%$ (NFE = 5), approaching advanced explicit GNN baselines while remaining efficient. On the larger ogbn-products benchmark, C-DEQ exhibits the same monotonic refinement behavior, reaching $76.55\%$ at NFE = 10, and it is substantially faster than IGNN-Solver (e.g., 3.85s vs. 9.30s at NFE = 10). Overall, these results indicate that C-DEQ effectively accelerates the convergence process into only a few function evaluations, yielding accuracy gains as inference steps increase. This confirms that our C-DEQ is a general and outstanding acceleration framework that scales to complex graph data and mitigates the inference bottleneck of iterative solvers on large-scale graphs.

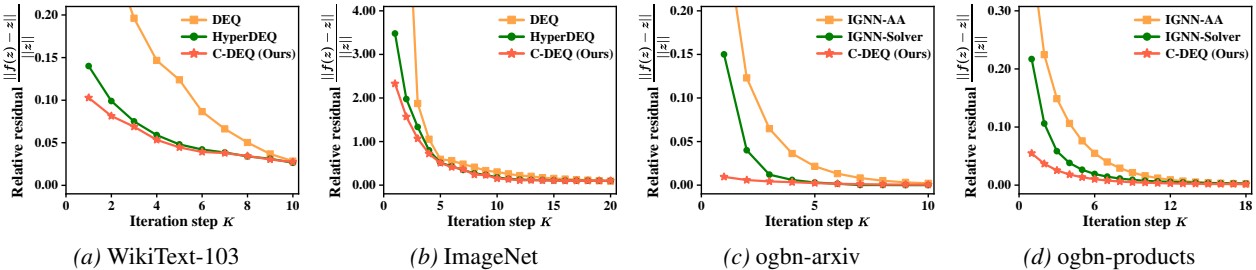

*(a) WikiText-103*  *(b) ImageNet*  *(c) ogbn-arxiv*  *(d) ogbn-products*

*Figure 2.* We visualize the residual trajectory across iteration steps $K$ on four benchmarks. Compared to standard DEQ (yellow) and previous acceleration methods like HyperDEQ/IGNN-Solver (green), our C-DEQ (red) consistently improves the convergence path and achieves lower residuals in fewer steps. Notably, on graph tasks (c, d) with theoretically well-posed backbones, C-DEQ reaches near-equilibrium almost instantaneously.

*Table 4.* Performance and Efficiency Comparison of the AA method against Picard and Broyden's methods on Various Datasets.

| Method | WikiText-103 (PPL↓ / s/batch) | ImageNet (Acc.↑ / s/batch) | ogbn-arxiv (Acc.↑ / s/batch) | ogbn-products (Acc.↑ / s/batch) |
|---|---|---|---|---|
| AA | 26.43 / 0.37 | 74.04 / 0.87 | 71.40 / 0.16 | 76.55 / 3.85 |
| Broyden | 26.97 / 0.43 | 71.82 / 1.05 | 67.83 / 0.17 | 74.25 / 4.23 |
| Picard | 28.68 / 0.33 | 69.46 / 0.85 | 61.13 / 0.14 | 63.91 / 3.56 |

## 4.2. Convergence Analysis

We present additional evidence on the convergence of C-DEQ in Figure 2, which illustrates the relative residual $\frac{||f(z)-z||}{||z||}$ ($|| \cdot ||$ denotes the $L_2$ norm) as a function of the iteration step $K$ across four diverse benchmarks. Across all DEQ variants, the relative residual decreases steadily as the iteration step increases. Vanilla DEQ converges the slowest, while HyperDEQ and C-DEQ accelerate convergence. Notably, C-DEQ reduces the residual more sharply in the first few steps than HyperDEQ, reaching a tighter approximation with fewer evaluations.

Another key observation is the distinct convergence behavior between graph-based tasks and other modalities: On the graph benchmarks (Figures 2c and 2d), the backbones are IGNN-based and explicitly designed to be well-posed; C-DEQ inherits this stability and therefore reaches a near-equilibrium state almost immediately, yielding a notably fast convergence profile than other DEQ variants. In contrast, for NLP and ImageNet (Figures 2a and 2b), the underlying architectures impose fewer theoretical constraints to ensure well-posedness, so while C-DEQ still converges faster than the baselines, the improvement is less. We further discuss how the well-posedness of the pretrained teacher affects the convergence behavior of C-DEQ in Appendix C.5.

## 4.3. Ablation Studies

In this section, we ablate three key design choices of C-DEQ: the trajectory solver, the AA-based history-dependent parameterization, and the loss coefficients $\lambda_1$ and $\lambda_2$. The corresponding results are summarized in Table 4, Figure 3 and Table 5, respectively. We report ablation results primarily on the WikiText-103 dataset, noting that consistent patterns are observed across other benchmarks.

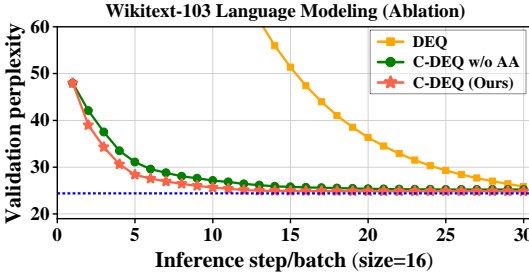

*Figure 3.* Ablative studies on C-DEQ. By explicitly leveraging solver history, C-DEQ (red) approaches the converged baseline (dotted) in only 5–6 steps, significantly faster than the non-AA variant (green) which requires nearly $2\times$ the steps.

Beyond the main ablations, Appendix B provides additional controlled studies on AA inductive bias, history size, and other hyperparameters. These include replacements for the AA module, sensitivity to the history window $m$, and sensitivity to $\gamma, \rho, \beta$ and the time mapping $t_k$. Together, these results demonstrate that our method is not only effective under the default setting, but also robust to alternative solver choices, architectural substitutions, and key hyperparameter variations.

**Ablation on Solver Choice.** Since the framework relies on solver-induced trajectories, the choice of trajectory solver may affect the quality of the intermediate targets. Besides Anderson acceleration (AA), alternatives such as Picard and Broyden's methods can also be used to generate teacher trajectories. We choose AA as the default solver due to its fast convergence and established effectiveness in DEQ

*Table 5.* Grid search on WikiText-103 (PPL↓) reveals that a hybrid consistency objective ($\lambda_1 = 0.8$) augmented with task-level regularization ($\lambda_2 = 0.05$) achieves the optimal performance.

| $\lambda_2$ \ $\lambda_1$ | 0 | 0.2 | 0.4 | 0.6 | 0.8 | 1 |
|---|---|---|---|---|---|---|
| 0 | 25.7 | 25.6 | 25.5 | 25.4 | 25.5 | 25.8 |
| 0.05 | 25.5 | 25.4 | 25.2 | 25.1 | **25.0** | 25.3 |

literature (Bai et al., 2021a; Baker et al., 2023), and further compare these alternatives in Table 4. All solver variants are evaluated under identical architectures, pretrained teachers, inference budgets, and optimization settings; only the trajectory solver is changed. The results show that AA provides the best performance-efficiency trade-off across tasks.

**Ablation on AA Parameterization.** We investigate the necessity of the proposed AA parameterization in Eq. (7), by comparing C-DEQ against a variant "C-DEQ w/o AA" which replaces the history-dependent update with a standard fixed-point step (i.e., relying solely on the current estimate). As illustrated in Figure 3, removing the structural prior of AA leads to a distinct degradation in inference efficiency. While both consistency-based models significantly outperform the standard DEQ baseline, the proposed C-DEQ (red) exhibits a substantially sharper convergence rate than its counterpart without AA (green). Specifically, although both variants start from the same performance level, our method approaches the converged baseline (dotted line) in only 5–6 steps; in contrast, the non-AA variant requires nearly $2\times$ the computational steps to achieve comparable perplexity. This confirms the necessity of the AA parameterization.

**Ablation on Loss Coefficients.** Table 5 investigates the interplay between the consistency balance $\lambda_1$ and the task regularizer $\lambda_2$ in Eq. (11). We follow the same experimental protocol, run the full training schedule with identical architecture and optimization settings, and report test perplexity under a fixed 20-step inference budget. With $\lambda_2 = 0$, we observe a convex performance trend for $\lambda_1$, where a balanced mixture (e.g., $\lambda_1 = 0.6$) significantly outperforms relying solely on global ($\lambda_1 = 1$) or local ($\lambda_1 = 0$) objectives. This confirms that both global and local consistency losses are therefore necessary to achieve accurate and stable few-step inference. Moreover, introducing the task-level constraint ($\lambda_2 = 0.05$) yields a consistent performance gain across all $\lambda_1$ settings. Overall, the optimal configuration is achieved at $\lambda_1 = 0.8$ and $\lambda_2 = 0.05$ with a PPL of 25.0, validating the effectiveness of our composite loss design.

## 5. Conclusion and Limitations

We proposed C-DEQ, a framework that accelerates DEQ inference by distilling along a fixed FP-ODE trajectory. By leveraging the AA structural prior and balancing global and local consistency, C-DEQs map intermediate states directly to the equilibrium, enabling few-step inference while retaining the benefits of iterative solving. Extensive experiments across vision, language, and graph tasks demonstrate that C-DEQs achieves consistent 2-20$\times$ accuracy improvements over implicit DEQs under the same few-step inference budget, effectively narrowing the efficiency gap between implicit DEQs and explicit networks.

**Limitations.** C-DEQ accelerates inference by distilling a pretrained DEQ teacher, but still requires training the original teacher model. The learned consistency map also depends on the stability of the teacher trajectory: less stable DEQ dynamics may require more accurate teacher solving or stronger regularization. Moreover, our FP-ODE view mainly serves as a trajectory-selection perspective; stronger theoretical guarantees and adaptive solver-induced trajectories are left for future work.

## Impact Statement

This paper presents work whose goal is to advance the field of Machine Learning. There are many potential societal consequences of our work, none of which we feel must be specifically highlighted here.

## Acknowledgements

Z. Ling would also like to acknowledge the National Natural Science Foundation of China (NSFC-62406119), the National Key Research and Development Program of China (No. 2025YFA1018600), the Natural Science Foundation of Hubei Province (2024AFB074), and the Fundamental Research Support Program of HUST (2025BRSXB0004). R. C. Qiu is partially supported in part by the National Natural Science Foundation of China (via NSFC-12141107), the Key Research and Development Program of Wuhan (2024050702030100), and the Key Research and Development Program of Guangxi (GuiKe-AB21196034).

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

# A. Experimental Details

## A.1. Baselines

To demonstrate the superiority of C-DEQ, we compare it against a range of traditional and learning-based DEQ acceleration methods across multiple domains. Specifically, on large-scale language datasets, we compare C-DEQ with several explicit state-of-the-art methods, including TCN, Gated ConvNet (GCNN), AWD-QRNN, and Transformer-XL, which represent the standard for high-performance sequence modeling. Furthermore, we include two representative DEQ-based Transformer models, namely the original DEQ and HyperDEQ, to evaluate our improvements over existing implicit sequence architectures.

For large-scale image classification tasks, the baselines cover a spectrum from foundational convolutional networks to modern multi-resolution designs. We include AlexNet and Inception-V2 as classic benchmarks, alongside HRNet-W18-C, which serves as a competitive explicit multi-scale baseline. To verify the efficiency of our consistency-based approach, we further compare C-DEQ with Single-stream DEQ and MDEQ, the latter being the state-of-the-art for multi-scale implicit modeling in computer vision.

For large-scale graph tasks, we benchmark C-DEQ against a comprehensive suite of Implicit Graph Neural Networks (IGNNs) that define node representations through fixed-point equilibrium equations. This includes the foundational IGNN, EIGNN, and MIGNN. Additionally, we compare against IGNN-Solver, a specialized method designed to accelerate the solver's convergence in graph-structured data, providing a direct comparison for our consistency-based acceleration framework.

## A.2. Datasets

**Language Modeling: WikiText-103.** Language modeling represents a particularly challenging case for implicit models because the sequential dependencies often necessitate a high number of iterations to achieve a stable hidden state. To assess the performance on sequence modeling, we utilize the large-scale WikiText-103 (WT103) corpus, which contains >100M words and a vocabulary size of >260K. Following standard protocols, we train our C-DEQ on sequences of length 60, and we use a validation sequence length of 150.

**Graph Benchmarks: OGBN.** We conduct experiments on two representative benchmarks from the Open Graph Benchmark (OGB), including: ogbn-arxiv: A directed citation network between Computer Science (CS) arXiv papers. Each node is associated with a 128-dimensional feature vector representing the average word embeddings of its title and abstract. The task is a 40-class classification problem to predict the subject area of each paper; and ogbn-products: A significantly larger undirected and unweighted graph representing an Amazon product co-purchasing network. Nodes represent individual products, and edges indicate that the products are frequently purchased together. The goal is to predict product categories across 47 top-level labels in a multi-class setup. These benchmarks are particularly suitable for evaluating C-DEQ, as IGNNs typically suffer from high latency during the iterative message-passing process required to reach equilibrium.

**Image Classification: ImageNet.** To evaluate the scalability of trajectory distillation in high-resolution vision tasks, we extend our evaluation to the ImageNet-1k dataset. This benchmark is a cornerstone of large-scale image classification, consisting of 1.2 million labeled training images across 1,000 classes. Following MDEQ (Bai et al., 2020), we evaluate on ImageNet-1k with $224 \times 224$ crops and report Top-1/Top-5 accuracy on the validation split. The high dimensionality of ImageNet features makes the iterative root-finding process of standard DEQs computationally expensive, providing a rigorous test for our distillation framework.

## A.3. Implementation Details

All models are implemented using PyTorch and trained on a machine equipped with an Intel(R) Xeon(R) Gold 6138 CPU @ 2.00 GHz and 8 NVIDIA RTX 3090 GPUs. For each baseline method, we strictly maintain the identical DEQ architectures and the same Anderson Acceleration configurations, such as the history size $m$ and convergence tolerance $\tau$, as specified in their respective original implementations. This ensures that the sampled iterative paths are consistent with the established performance of these models. Further details regarding the specific hyperparameter configurations for all baselines and C-DEQ across datasets are provided in Table 6.

*Table 6.* **Hyperparameter configurations for all compared methods across datasets.** For all baselines, we follow the hyperparameter settings reported in their original papers or official implementations.

| Datasets | Model | # Params | Opt. | Learning Rate | Weight Decay | Dropout |
|---|---|---|---|---|---|---|
| WikiText-103 | Generic TCN (Bai, 2018) | 150M | SGD | 4.0 | 0 | 0.45 |
| | GCNN (Dauphin et al., 2017) | 230M | NAG | 1.0 | 5e-6 | 0.2 |
| | AWD-QRNN (Merity et al., 2018) | 159M | ASGD | 30 | 1.2e-6 | 0.1 |
| | Transformer-XL (Dai et al., 2019) | 165M | Adam | 2.5e-4 | 0 | 0.1 |
| | DEQ (Bai et al., 2019) | 168M | Adam | 3e-4 | 0 | 0.05 |
| | HyperDEQ (Bai et al., 2021a) | 168M | Adam | 3e-4 | 0 | 0.05 |
| | **C-DEQ (Ours)** | 170M | Adam | 3e-4 | 0 | 0.05 |
| ImageNet | AlexNet (Krizhevsky et al., 2012) | 61M | SGD | 0.01 | 5e-4 | 0.5 |
| | Inception-V2 (Ioffe & Szegedy, 2015) | 12M | SGD | 0.045 | 4e-5 | 0 |
| | HRNet-W18-C (Wang et al., 2020) | 21M | SGD | 0.1 | 1e-4 | 0 |
| | DEQ (Bai et al., 2019) | 18M | SGD | 0.05 | 5e-5 | 0 |
| | HyperDEQ (Bai et al., 2021a) | 18M | SGD | 0.05 | 5e-5 | 0 |
| | MDEQ (Bai et al., 2020) | 18M | SGD | 0.05 | 5e-5 | 0 |
| | **C-DEQ (Ours)** | 18M | SGD | 0.05 | 5e-5 | 0 |
| Ogbn-Arxiv | GCN (Kipf & Welling, 2017) | 1.46M | Adam | 0.01 | 5e-4 | 0.5 |
| | GAT (Veličković et al., 2018) | 1.44M | Adam | 5e-3 | 5e-4 | 0.8 |
| | GCNII (Chen et al., 2020) | 2.15M | Adam | 0.01 | 5e-4 | 0.5 |
| | GPM (Wang et al., 2025) | 1.40M | Adam | 0.01 | 5e-4 | 0 |
| | IGNN (Gu et al., 2020) | 0.15M | Adam | 0.01 | 1e-3 | 0.5 |
| | EIGNN (Liu et al., 2021) | 0.15M | Adam | 0.8 | 5e-6 | 0.5 |
| | MIGNN (Baker et al., 2023) | 0.16M | Adam | 8e-3 | 5e-4 | 0.4 |
| | IGNN-Solver (Lin et al., 2024) | 0.16M | Adam | 1e-3 | 5e-4 | 0.5 |
| | **C-DEQ (Ours)** | 0.16M | Adam | 0.01 | 5e-4 | 0.5 |
| Ogbn-Products | GCN (Kipf & Welling, 2017) | 2.55M | Adam | 0.01 | 5e-4 | 0.5 |
| | GAT (Veličković et al., 2018) | 2.62M | Adam | 5e-3 | 5e-4 | 0.8 |
| | GCNII (Chen et al., 2020) | 3.79M | Adam | 0.01 | 5e-4 | 0.5 |
| | GPM (Wang et al., 2025) | 2.45M | Adam | 0.01 | 5e-4 | 0 |
| | IGNN (Gu et al., 2020) | 0.17M | Adam | 0.01 | 1e-3 | 0.5 |
| | EIGNN (Liu et al., 2021) | 0.17M | Adam | 0.8 | 5e-6 | 0.5 |
| | MIGNN (Baker et al., 2023) | 0.18M | Adam | 8e-3 | 5e-4 | 0.4 |
| | IGNN-Solver (Lin et al., 2024) | 0.18M | Adam | 1e-3 | 5e-4 | 0.5 |
| | **C-DEQ (Ours)** | 0.18M | Adam | 0.01 | 5e-4 | 0.5 |

## A.4. Detailed Efficiency Analysis.

**Comparison on Training Time.** We directly use the weights of a pretrained DEQ as initialization, following standard practice in prior work such as HyperDEQ and Skip-DEQ. We additionally compare the total wall-clock distillation training time under the same setting in Table 7. The results show that the training time required by C-DEQ accounts for only a small fraction of that of DEQ.

*Table 7.* Additional Distillation Time of C-DEQ Compared with Teacher DEQ Pretraining Time.

| Training time | C-DEQ | DEQ |
|---|---|---|
| WikiText-103 | 2.6h | 36.2h |
| ImageNet | 4.5h | 57.4h |

**Training overhead.** The overhead from trajectory caching and distillation is modest. In Table 8, "traj." denotes a cached trajectory batch under our experimental batch setting, rather than the full dataset or a single sample. "Storage" reports the footprint of storing all cached states in the trajectory batch, "Generation" measures the time required to produce the teacher trajectory, "# Points" represent the number of stored solver states, "Training Time" is the wall-clock cost of one distillation update batch, and "Peak GPU Mem." is the maximum GPU memory used during distillation. Across benchmarks, trajectory generation takes only 0.97–2.61s per cached trajectory batch, with storage ranging from 173.1MB to 757.1MB. For ImageNet, a 50-point trajectory batch requires 757.1MB and 2.61s to generate, while each distillation update batch takes 4.93s with 20.94GB peak GPU memory. Together with Table 7, these results show that C-DEQ introduces only a modest offline cost while substantially reducing inference latency.

*Table 8.* Training Overhead of Trajectory Caching and Distillation in C-DEQ.

|  | Storage (MB/traj. batch) | Generation (s/traj. batch) | # Points/traj. | Training Time (s/batch) | Peak GPU Mem. (GB) |
|---|---|---|---|---|---|
| WikiText-103 | 337.2 | 1.44 | 40 | 3.27 | 17.02 |
| ImageNet | 757.1 | 2.61 | 50 | 4.93 | 20.94 |
| ogbn-arxiv | 173.1 | 0.97 | 30 | 1.21 | 12.90 |
| ogbn-products | 313.9 | 1.34 | 40 | 1.80 | 15.49 |

## B. More Experiments

**Ablation on AA.** To show that the observed performance gain is derived from the specific inductive bias of the AA mathematical formulation, rather than simply from the availability of additional historical information, we conduct an ablation replacing the AA module with generic MLP and RNN (Table 9). Both MLP and RNN fail to match the convergence speed and accuracy of the AA-based design. These results indicate that simply providing historical information is insufficient; the structured inductive bias of AA is the key factor behind performance gains.

*Table 9.* Performance and Efficiency of AA Module Ablations. We compare the AA module against: (i) an MLP baseline (two-layer bottleneck, $\mathbb{R}^{4d} \to \mathbb{R}^{2d} \to \mathbb{R}^d$) and (ii) an RNN variant (GRU with hidden size $d$), both of which take the concatenated current and previous states as input.

|  | WikiText-103 | | ImageNet | | ogbn-arxiv | |
|---|---|---|---|---|---|---|
|  | PPL | Eff. (s/batch) | Top1 Acc. | Eff. (s/batch) | Acc | Eff. (s/batch) |
| AA | 26.43 | 0.37 | 74.04 | 0.87 | 71.40 | 0.16 |
| MLP | 38.93 | 0.69 | 63.49 | 1.17 | 66.81 | 0.28 |
| RNN | 47.08 | 0.74 | 59.86 | 1.26 | 64.53 | 0.27 |

**Sensitivity on $m$.** We conduct an ablation study in Table 10 to evaluate how $m$ affects performance and efficiency. Increasing $m$ yields only slight accuracy gains without clear saturation, but incurs much higher inference latency. A small $m$ provides the best accuracy–efficiency trade-off.

*Table 10.* Sensitivity Analysis of History Window Size $m$ on Various Datasets.

|  | WikiText-103 | | ImageNet | | ogbn-arxiv | |
|---|---|---|---|---|---|---|
|  | PPL | Eff. (s/batch) | Top1 Acc. | Eff. (s/batch) | Acc | Eff. (s/batch) |
| $m = 1$ | 26.43 | 0.37 | 74.04 | 0.87 | 71.40 | 0.16 |
| $m = 3$ | 26.23 | 0.89 | 74.81 | 1.28 | 71.66 | 0.29 |
| $m = 5$ | 26.04 | 1.72 | 75.03 | 1.95 | 71.71 | 0.43 |
| $m = 7$ | 25.93 | 2.96 | 75.09 | 3.02 | 71.73 | 0.85 |

**Ablation studies on $\gamma$, $\rho$ and $\beta$.** We provide default hyperparameter settings and additional ablation studies for all key parameters. We determine $\gamma$, $\rho$ and $\beta$ via small-scale grid search. Specifically, we set $\gamma = 2$ and $\rho = 0.1$ for trajectory sampling, and $\beta = 0.9$ for multi-step inference. In Algorithm 4, we follow the standard CLLM (Kou et al., 2024) setup with $p_{aug} = 0.1$, $k_{min} = 3$, and $k_{tail} = 2$. To assess sensitivity, we conduct experiments on WikiText-103 varying $\gamma$, $\rho$, as well as $\beta$. Empirical results (summarized in Tables 11 and 12) show that within a reasonable hyperparameter range, C-DEQ consistently maintains its performance advantage without requiring extensive tuning.

**Ablation studies on $t_k$.** We adopt the exponential mapping in $t_k$ following CM setup. In typical fixed-point iterations, the most significant state updates and residual reductions occur during the early steps, followed by minor asymptotic refinements later. An exponential schedule naturally allocates higher temporal resolution to these critical early dynamics. To empirically validate this, we evaluated alternative schedules and validate it against linear and cosine mappings on WikiText-103, keeping all other settings identical. As shown in Table 13, it outperforms linear and cosine mappings.

*Table 11.* Ablation study on hyperparameters $\gamma$ and $\rho$.

| PPL↓ | $\gamma = 0.5$ | $\gamma = 1.0$ | $\gamma = 2.0$ | $\gamma = 5.0$ |
|---|---|---|---|---|
| $\rho = 0.05$ | 93.57 | 35.29 | 27.69 | 51.52 |
| $\rho = 0.1$ | 89.08 | 34.24 | 26.43 | 50.18 |
| $\rho = 0.2$ | 97.23 | 36.03 | 30.45 | 53.29 |

*Table 12.* Ablation study on hyperparameter $\beta$.

| $\beta$ | 0.5 | 0.7 | 0.9 | 0.95 |
|---|---|---|---|---|
| PPL | 29.28 | 28.19 | 26.43 | 26.72 |

## C. Discussion

### C.1. Anderson Acceleration Solver and FP-ODE

To recover the equilibrium state in implicit networks efficiently, Anderson Acceleration (AA) (Anderson, 1965) is widely employed as a robust fixed-point solver (Bai et al., 2019; 2020; Baker et al., 2023). Specifically, for an implicit layer defined by $z = f_\theta(z, x)$, the goal is to identify the fixed point $z^\star$ that satisfies Eq. (1). This is equivalent to solving the root-finding problem $F_\theta(z; x) = f_\theta(z, x) - z = 0$. As a quasi-Newton method, AA effectively accelerates convergence by leveraging the history of previous iterates, often demonstrating super-linear convergence rates in practice (Walker & Ni, 2011).

As summarized in Algorithm 3, the core mechanism of AA involves constructing the next approximate solution $z_{k+1}$ as a linear combination of the past $m$ iterates and their function evaluations. Specifically, it seeks coefficients $\{\alpha_i\}_{i=0}^{m-1}$ that minimize the norm of the residual for the corresponding linear combination. Let $R_k = f_\theta(z_k, x) - z_k$ be the residual at step $k$. AA solves a constrained least-squares problem: $\min_\alpha \| \sum_{i=0}^{m_k} \alpha_k^i R_{k-m_k+i} \|_2$ subject to $\sum \alpha_k^i = 1$. By introducing a mixing parameter $\beta \in (0, 1]$ to control the step size and stabilize the update, the iteration step is explicitly formulated as:

$$z_{k+1} = \beta \sum_{i=0}^{m_k} \alpha_k^i f_\theta(z_{k-m_k+i}) + (1 - \beta) \sum_{i=0}^{m_k} \alpha_k^i z_{k-m_k+i} \tag{12}$$

By utilizing this historical information, AA bypasses the limitations of Picard iterations and often achieves super-linear convergence. Furthermore, since the backward pass via implicit differentiation depends only on the final state $z^\star$ and is independent of the internal solver trajectory, DEQs maintain constant memory consumption during training (Bai et al., 2019). However, the iterative nature of AA still imposes a sequential bottleneck during inference, which motivates our transition toward trajectory distillation.

We use the FP-ODE view as a trajectory-selection and distillation perspective rather than as a complete convergence theory for arbitrary DEQs. When the pretrained DEQ operator is well-posed, the induced fixed-point flow shares the same equilibrium as the DEQ equation. In practice, we instantiate this trajectory using AA, which produces a solver-informed discrete path toward the same fixed point.

### C.2. Augmented Trajectory Sampling

Consistency distillation in C-DEQ requires training data in the form of solver trajectories rather than independent input–output pairs. Given a pretrained DEQ model, the initial state $z_0$ and solver $\mathcal{S}$, we generate trajectory data by running a fixed-point solver and recording its intermediate states along the iteration. The teacher solver runs for $K$ iterations, producing

*Table 13.* Ablation Study on Mapping Function $t_k$ across Various Datasets.

| | $t_k = \epsilon + \frac{k}{K}(T - \epsilon)$ (Linear mapping) | $t_k = \epsilon + (1 - \cos(\frac{k\pi}{2K}))(T - \epsilon)$ (Cosine mapping) | $t_k = \epsilon + (1 - e^{-\rho k})(T - \epsilon)$ (Ours) |
|---|---|---|---|
| WikiText-103 | 31.91 | 29.77 | 26.43 |
| ImageNet | 69.16 | 71.50 | 74.04 |
| ogbn-arxiv | 67.62 | 68.04 | 71.40 |
| ogbn-products | 70.45 | 72.36 | 76.55 |

---

**Algorithm 3** Anderson Acceleration Solver

---

1: **Input:** fixed-point function $f_{\boldsymbol{\theta}} : \mathbb{R}^n \to \mathbb{R}^n$, max storage size $m$, the past residuals $\boldsymbol{R}_k = [F_{\boldsymbol{\theta}}(\boldsymbol{z}_{k-m_k}), \ldots, F_{\boldsymbol{\theta}}(\boldsymbol{z}_k)]$, residual control parameter $\beta$
2: Set initial point value $\boldsymbol{z}_0 \in \mathbb{R}^n$
3: **for** $k = 0, \ldots, K - 1$ **do**
4:     1) Set $m_k = \min\{m, k\}$
5:     2) Solve weights $\boldsymbol{\alpha}_k = \arg\min_{\boldsymbol{\alpha} \in \mathbb{R}_{m_k+1}} \|\boldsymbol{R}_k \boldsymbol{\alpha}\|_F$ for the past $m_k$ Anderson steps, s.t. $\mathbf{1}^\top \boldsymbol{\alpha}_k = 1$
6:     3) $\boldsymbol{z}_{k+1} = \beta \sum_{i=0}^{m_k} \boldsymbol{\alpha}_k^i f_{\boldsymbol{\theta}}(\boldsymbol{z}_{k-m_k+i}) + (1 - \beta) \sum_{i=0}^{m_k} \boldsymbol{\alpha}_k^i \boldsymbol{z}_{k-m_k+i}$
7: **end for**

---

states $\{\boldsymbol{z}_k\}_{k=0}^K$ with $\boldsymbol{z}_K \approx \boldsymbol{z}^\star$. Each solver step naturally corresponds to a progression in solver time, yielding a temporally ordered sequence of states that can be directly used for consistency distillation. In practice, sampled solver trajectories are generated once and cached, after which intermediate states are reused as training samples throughout distillation. This allows each intermediate solver state to be treated as separate supervised pairs conditioned on the same input $\boldsymbol{x}$, improving sample reuse and stabilizing optimization.

To enhance the learning and generalization capabilities of C-DEQ, we introduce a data augmentation scheme that randomly replaces intermediate points in the trajectory with the final equilibrium state $\boldsymbol{z}^\star$. Specifically, for any given point $\boldsymbol{z}_k$ in the sequence during sampling (excluding the first point and the last few points to preserve the core dynamics), we replace it with the trajectory endpoint $\boldsymbol{z}_K$ with a certain probability $p_{\text{aug}}$:

$$\boldsymbol{z}_k \leftarrow \boldsymbol{z}_K, \quad \text{with probability } p_{\text{aug}}. \tag{13}$$

This strategy exposes the model to the equilibrium state early, encouraging $g_{\boldsymbol{\theta}}$ to learn the fixed-point property and "short-cut" the trajectory toward $\boldsymbol{z}^\star$ during training; and acts as a stochastic regularization that prevents over-fitting to specific paths. The completed trajectory sampling algorithm is shown in Algorithm 4.

---

**Algorithm 4** Augmented Trajectory Sampling for Consistency Distillation

---

**Require:** Pretrained DEQ model $f_{\boldsymbol{\theta}}$, AA solver $\mathcal{S}^{\text{AA}}$, input $\boldsymbol{x}$, maximum iteration $K$, augmentation probability $p_{\text{aug}}$, exclusion parameters $k_{\min}$ and $k_{\text{tail}}$
**Ensure:** Cached augmented trajectory $\{\tilde{\boldsymbol{z}}_k\}_{k=0}^K$ and endpoint $\boldsymbol{z}^\star$
1: Initialization $\boldsymbol{z}_0 = 0$
2: **for** $k = 0$ to $K - 1$ **do**
3:     $\boldsymbol{z}_{k+1} = \mathcal{S}^{\text{AA}}(\{\boldsymbol{z}_{k-m_k+i}\}_{i=0}^{m_k}; f_{\boldsymbol{\theta}}, \boldsymbol{x})$ via Eq. (3)
4: **end for**
5: Set endpoint $\boldsymbol{z}^\star \leftarrow \boldsymbol{z}_K$
6: Initialize augmented trajectory $\tilde{\boldsymbol{z}}_k \leftarrow \boldsymbol{z}_k$ for all $k$
7: **for** $k = k_{\min}$ to $K - k_{\text{tail}}$ **do**
8:     Draw $u \sim \text{Uniform}(0, 1)$
9:     **if** $u < p_{\text{aug}}$ **then**
10:        $\tilde{\boldsymbol{z}}_k \leftarrow \boldsymbol{z}^\star$
11:    **end if**
12: **end for**
13: **return** $\{\tilde{\boldsymbol{z}}_k\}_{k=0}^K$

---

### C.3. Construction of Solver-Time Schedules

Here we provide technical details on the construction of the solver-time schedules used for mapping from the solver iteration index $k$ to virtual time $t_k \in [\epsilon, T]$ in trajectory sampling (Section 3.2) and multi-step inference (Section 3.3).

**Solver-Time Schedules for Trajectory Sampling.**    During the consistency distillation phase, solver states are indexed by the discrete iteration counter $k \in \{0, 1, \ldots, K\}$. To map this progression into the bounded learning-time interval $[\epsilon, T]$, we introduce the normalized ratio $1 - e^{-\rho k} \in [0, 1)$ (utilized in Eq. (6)), where $\rho > 0$ is a hyperparameter controlling the

compression rate. This exponential form allocates higher temporal resolution to early solver iterations where state transitions are most rapid (the transient phase), while progressively compressing later iterates as they approach the stable manifold.

**Solver-Time Schedules for Multi-Step Inference.** In inference stage, a trained C-DEQ $h_\phi$ applies the consistency function $h_\phi$ at a sequence of $J$ progressively later solver times. Since the model is trained to be consistent across the entire time manifold, we are not restricted to the training schedule, rather select a set of monotonically increasing ratios $\{1 - \beta^j\}_{j=1}^J$ in practice. Therefore, the time mapping can be defined as $t_j = \epsilon + (1 - \beta^j)(T - \epsilon)$, from inference index $j$ to virtual time $t_j \in [\epsilon, T)$.

## C.4. Auxiliary Task-Level Regularization

In addition to the primary trajectory-based distillation objective, we incorporate a lightweight task-level loss acting as a task-relevant regularizer. Unlike standard supervision, this term is not intended to drive the optimization of the solver dynamics directly, but rather to ensure the intermediate states retain task-relevant information, thereby improving training stability.

Concretely, let $y$ denote the ground-truth label (e.g., the next token index in NLP or class label in classification), $h_\phi(z_t, t, x)$ be the student consistency function that takes a solver state $z_t \in \mathbb{R}^d$ at time $t$ and outputs a refined state, $h(\cdot)$ be the frozen task prediction head of the pretrained teacher model, which maps a state vector to task logits. We define the task-level regularization loss as:

$$\mathcal{L}_{\text{task}} = \mathbb{E}_{t \sim p(t)} \, \text{CE}(h(h_\phi(z_t, t)), \, y) \,, \tag{14}$$

where $\text{CE}(\cdot, \cdot)$ is the cross-entropy loss between the predicted logits and the label $y$, and $p(t)$ is a sampling distribution over solver times (consistent with our time parameterization in Eq. (6)). In practice, we restrict this regularization to the early phase of the solver trajectory (where $t$ is small) and assign it a significantly lower weight. This ensures it serves only as a mild constraint without compromising the solver-centric nature of C-DEQ.

## C.5. Impact of Teacher Well-Posedness on Convergence

Figure 2 shows near-instantaneous convergence on graph tasks but notably slower refinement on NLP and ImageNet. The differences are not caused by data domain, but by the intrinsic properties of the teacher DEQ models. Since C-DEQ is initialized from the teacher DEQ and trained via trajectory distillation, it inherently inherits structural properties of the teacher, including its degree of well-posedness. In graph tasks, the teacher IGNN is explicitly designed to ensure strong well-posedness, which leads to the observed near-instant convergence. In contrast, teacher DEQs used in NLP and Vision do not always enforce such strict properties during pretraining, resulting in slower convergence behaviors.

