# OpenReview forum: "Consistency Deep Equilibrium Models"
_ICML.cc/2026/Conference — ICML 2026 regular_

### Official Review · Reviewer_n8bh · 2026-03-03

**Soundness:** 2
**Presentation:** 2
**Significance:** 3
**Originality:** 2
**Overall Recommendation:** 3
**Confidence:** 3

**Summary:**

The paper proposes a novel consistency-based approach to DEQs with the intent to accelerate inference. The approach combines multiple methods, including consistency parametrization, Anderson acceleration, transcription into a fixed-point ODE, time mapping etc. At page 5, Algorithm 1 is for inference and Algorithm 2 is for training. The proposed approach is evaluated on 3 modalities of benchmarks: natural language processing, computer vision, and graph learning.

**Compliance With Llm Reviewing Policy:**

Affirmed.

**Key Questions For Authors:**

Please clarify the connection between DEQ, AA and consistency models through the FP-ODE.
Please clarify why the FP-ODE had indeed a unque fixed point and the solutions converge to it.

**Limitations:**

While I appreciated the numerical results, I did not understand sufficiently clearly the approach nor did it appear to me that the approach is rigorous.

**Strengths And Weaknesses:**

1. Minor comment: As much as I like equilibrium and implicit models, I am not sure that it is widely established that DEQs have demonstrated remarkable performance and, specifically, outperformance with respect to standard feedforward architectures.
2. In the Methods, section 3.1, an FP-ODE is introduced in equation (2). The document provides no guarantee that this ode will converge to the solution to the fixed point problem in equation (1). This is a critical point as otherwise the entire consistency approach is flawed.
3. In Section 3.2, the paper introduced Anderson acceleration (AA). Apparently this is for the function f_\theta and not for the FP-ODE. It is not clear why the AA for f_\theta relates to the solutions of the FP-ODE.
4. According to point 3, it is not clear what is being displayed in Figure 1: in the left panel I understand that I need to select an ODE to have a unique path to consistently map --- but then it is not clear that the trajectory of the ODE is the one that is being approximated by the consistency maps.
5. Overall, the architecture feels cumbersome: we start with a single function, and then, instead of devising a quick way to solve it, we introduce a number of auxliary constructions.
6. Anderson acceleration is known to converge in some cases -- the document does not even hint at whether the setup in this paper matches any of those cases (e.g., the map f_\theta is a contraction or so)

---

> ### Author Rebuttal · Authors · 2026-03-31
>
> We greatly appreciate your constructive comments and address your concerns below.
>
> **On the Role and Convergence of FP-ODE:** To clarify, the continuous FP-ODE introduced in Eq. 2 is guaranteed to converge to the exact FP solution of Eq. 1 under standard well-posedness conditions. We address this from three perspectives:
> 1. Algebraic Perspective: If Eq. 1 is well-posed and has a unique fixed point, the FP-ODE reaches a steady state exactly when its derivative vanishes ($dz_t/dt=0$). Algebraically, this condition is identical to the original fixed-point equation.
> 2. Dynamical Systems Perspective: Established dynamical systems theory [1] dictates that if operator $f_\theta$ satisfies DEQ well-posedness conditions, the corresponding continuous ODE is *globally asymptotically stable* and will unconditionally converge to this unique fixed point. We kindly note that similar theoretical conclusions are also well introduced in [2], which is discussed in the related-work sec.
> 3. Empirical Validation: Empirically, as observed in Fig. 2, our C-DEQ exhibits a consistent convergence trend across all evaluated datasets. For example, on ogbn-arxiv, C-DEQ successfully reaches the fixed point in just 3 steps.
>
> In the revised manuscript, we will explicitly state the convergence conditions and emphasize that FP-ODE equilibrium is guaranteed to match the DEQ fixed point.
>
> **Connections between AA and FP-ODE:**
> We would like to clarify that, under standard contractivity assumptions on the residual mapping, $r(z):=f_\theta(z,x)-z$ and with a suitable choice of step size $\beta$, AA effectively accelerates the discrete FP-ODE trajectory, and the iterates converge to the FP-ODE solution. In other words, applying AA to $f_\theta$ does not break the alignment with the FP-ODE vector field; instead, it optimizes the trajectory along the same flow. Specifically:
>
> AA can be interpreted as a multi-step method applied to the discrete FP-ODE trajectory. Denoting the residuals of previous iterates as $r_j=f_\theta(z_j,x)-z_j$, the update of Eqs. 3-4 reads:
> $$z_{k+1}=\sum_{i=0}^{m_k}\alpha_iz_{k-m+i}+\beta\sum_{i=0}^{m_k}\alpha_ir_{k-m_k+i}$$
> where ${\alpha_i}$ minimizes the norm of the residual combination. Note that in the first term, the trajectory averaging does not introduce new directions; and in the second term, each residual $r_j$ points along the FP-ODE vector field defined in Eq.(2), so the iterates remain aligned with the FP-ODE flow.
>
> Under the above conditions, the optimized residual combination tends to zero, ensuring that AA converges to FP-ODE equilibrium. We will incorporate this discussion, emphasizing the sufficient conditions for convergence and the connection between AA and FP-ODE solution.
>
> **Clarification of Figure 1:** The trajectory of FP-ODE shown in the left panel is the one being approximated by the consistency maps. Fig. 1 plots a continuous FP-ODE trajectory, while our consistency map learns the corresponding discrete trajectory produced by AA. As explained above, the discrete trajectory remains aligned with the FP-ODE flow, ensuring that the consistency map approximates the FP-ODE trajectory faithfully.
>
> **Connections between DEQ, FP-ODE, AA, and CMs:**
> 1. DEQ to FP-ODE (Continuous Perspective): As noted in Sec. 3.1, the root-finding for DEQ (Eq. 1) is equivalent to finding the equilibrium of FP-ODE. The FP-ODE provides a continuous, deterministic trajectory, which is required for CD.
> 2. FP-ODE to AA (Discrete Sampling): As we explain above, AA provides a discrete trajectory aligns that with the continuous dynamics, enabling CD training in practice.
> 3. AA to CMs (Direct Mapping): Finally, with the discrete trajectory sampled by AA, our C-DEQ is trained to map intermediate states directly to the final equilibrium.
>
> **AA Convergence:** We will explicitly state that we assume that the pre-trained DEQ is well-posed, i.e., the map is a contraction. Under this assumption, applying AA to the discrete FP-ODE iterates is guaranteed to converge.
>
> **A Unified Framework:**
> We respectfully clarify that our architecture is not a loose aggregation of components, but a *coherent framework grounded in a unified FP-ODE perspective*. Each component has a specific role within this principled view:
> - FP-ODE defines the continuous trajectory of the solution.
> - AA provides stable, informative discretization aligned with FP-ODE and makes CD tractable in practice.
> - CD enforces direct mapping that shortcuts iterative solving.
>
> These components form a structured decomposition of the core problem: constructing continuous trajectories via the ODE view, discretizing them with AA, and distilling them into an efficient solver. Comprehensive empirical results and ablation studies across domains confirm that each component is both principled and necessary.
>
> Reference:
>
> [1] Ortega., et al. Iterative Solution of Nonlinear Equations in Several Variables. SIAM 1970.
>
> [2] Ding, S., et al. Two sides of the same coin. In *NeurIPS* 2023.

---

> > ### Author Rebuttal · Reviewer_n8bh · 2026-04-03
> >
> > The reply refers to "standard well-posedness" and, in a different point,
> > refers to an entire book [1] (which is really not about dynamical systems
> > theory) and to reference [2] (which is really not about the convergence of
> > ODEs).  I find the statement "In the revised manuscript, we will explicitly
> > state the convergence conditions and ..."  not convincing and I do not
> > understand why the rebuttal does not already list what the conditions would
> > be. On the matter of Anderson Acceleration, while the rebuttal presents a
> > review of its definition -- I believe there is still a gap between
> > theoretical analysis and practical effectiveness and I note that, again the
> > rebuttal does not cite established references.  Given this analysis, I
> > maintain my evaluation.

---

> > > ### Author Response · Authors · 2026-04-07
> > >
> > > We thank the reviewer for the thoughtful feedback. Below, we explicitly define the convergence conditions and clarify the theoretical foundation of our work.
> > >
> > >
> > > **Explicit condition:** As requested, the convergence condition of the fixed-point iteration in Eq. (1) is
> > >
> > > $$\rho(J) < 1, \quad J = \left. \nabla_z f_\theta \right|_{z=z^*}, $$
> > >
> > > where $\rho(\cdot)$ denotes the spectral radius and   $f_\theta(\cdot, x)$ is assumed differentiable. Based on this condition, it follows that $Re(\lambda_i (J)- 1) < 0$, i.e., the eigenvalues of the FP-ODE's Jacobian $J-I$ have strictly negative real parts, which ensures the continuous stability of the FP-ODE; that is, $\forall z(0) \in \mathbb{R}^n, \lim_{t \to \infty} \|z(t) - z^*\| = 0$, thereby guaranteeing convergence to a unique equilibrium. Moreover, note that the FP-ODE reaches equilibrium when the dynamics vanish ($\frac{dz}{dt}  = 0$) which coincides with the fixed-point equation.  Therefore, the FP-ODE in Eq. (2) will converge to the same fixed point as Eq. (1). In our work, we assume this standard well-posedness condition holds for the pretrained teacher DEQ, which is a common assumption in prior DEQ literature. Our method inherits the same architecture. We will incorporate the condition explicitly into the revised manuscript.
> > >
> > > We also acknowledge the reviewer’s concern regarding citation clarity. In the revision, we will explicitly cite the precise contraction mapping result (e.g., the relevant theorem (Theorem 10.1.3 of [1])) rather than referring broadly to general sources.
> > >
> > >
> > > **Clarification on AA and ODE Connection:** We agree that there is generally a gap between theoretical analysis and practical behavior for AA, and we will strengthen the discussion with appropriate, established references in the revision.
> > >
> > > Importantly, our intention was not to associate AA with a specific continuous-time ODE. As you correctly pointed out, different solvers  may induce different implicit dynamics while converging to the same fixed point. Our initial presentation of Eq. (2) as a specific FP-ODE discretized by AA was therefore misleading.
> > >
> > > Instead, Eq. (2) should be understood as a conceptual illustration. Our framework does not rely on the specific ODE form, but on the well-defined (path-dependent and continuous) trajectory that converges to the fixed point. This perspective is sufficient to motivate our CD framework and is consistent with the behavior of practical DEQ solvers.
> > >
> > > To avoid this confusion, we will 1) explicitly label Eq. (2) as a "conceptual FP-ODE illustration" and replace it with a general formulation $$\frac{d\mathbf{z}(t)}{dt} = G(\mathbf{z}_t, t; \boldsymbol{\theta})$$
> > > where $G$ is only a conceptual function that represents the solver-induced dynamics (which may be history-dependent); and 2) add a discussion on the specific formulation of $G$ for different solvers. For Picard and Quasi-Newton methods, their corresponding FP-ODE formulations are well-defined in (Section 2.1 in [2]) and (Section 4 in [3]). For AA, its dynamics can be interpreted as an approximation to quasi-Newton dynamics, which has been extensively discussed in well-established literature (Section 3 in [4] and Section 2.5 in [5]). We also note that recent works [6] have begun to study continuous-time limits of AA, often resulting in higher-order or memory-dependent dynamical systems. Our choice of AA is based on empirical results (please see the supplemental experiment in Table C at https://anonymous.4open.science/r/ICML2026_Rebuttal_table_a-4640/Results.md).
> > >
> > > We will include these finding in the revised version of the manuscript.
> > >
> > >
> > > [1] Ortega, J. M., & Rheinboldt, W. C. (2000). Iterative solution of nonlinear equations in several variables. Society for Industrial and Applied Mathematics.
> > >
> > > [2] Scieur, D., Roulet, V., Bach, F., & d'Aspremont, A. (2017). Integration methods and optimization algorithms. Advances in Neural Information Processing Systems, 30.
> > >
> > > [3] Ding, S., Cui, T., Wang, J., & Shi, Y. (2023). Two sides of the same coin: Bridging deep equilibrium models and neural ODEs via homotopy continuation. Advances in Neural Information Processing Systems, 36, 28053-28071.
> > >
> > > [4] Walker, H. F., & Ni, P. (2011). Anderson acceleration for fixed-point iterations. SIAM Journal on Numerical Analysis, 49(4), 1715-1735.
> > >
> > > [5] Fang, H. R., & Saad, Y. (2009). Two classes of multisecant methods for nonlinear acceleration. Numerical linear algebra with applications, 16(3), 197-221.
> > >
> > > [6] Chen, K., Jiang, Y., & Vuik, K. (2025). The Dynamical Anatomy of Anderson Acceleration: From Adaptive Momentum to Variable-Mass ODEs. arXiv preprint arXiv:2512.21269.

---

### Official Review · Reviewer_U1DK · 2026-03-04

**Soundness:** 3
**Presentation:** 3
**Significance:** 3
**Originality:** 3
**Overall Recommendation:** 4
**Confidence:** 3

**Summary:**

The paper proposes C-DEQ, which adapts consistency distillation (originally developed for accelerating diffusion models) to Deep Equilibrium Models (DEQs), addressing their slow iterative inference. Since DEQs are path-independent (the fixed point is well-defined, but intermediate states depend on initialization and solver choice), the authors fix the initial state and solver to define a deterministic Fixed-Point ODE (FP-ODE) trajectory suitable for distillation. A student model whose parameterization embeds Anderson Acceleration's update structure is then trained to map intermediate solver states directly to the equilibrium in a small number of steps. Experiments on WikiText-103, ImageNet, and OGB benchmarks show consistent accuracy gains over DEQ and HyperDEQ under matched NFEs, with performance improving as inference steps increase.

**Compliance With Llm Reviewing Policy:**

Affirmed.

**Final Justification:**

The proposed approach is technically sound and well-motivated, adapting consistency distillation to DEQs through a deterministic trajectory formulation, with consistent empirical improvements across NLP, vision, and graph benchmarks. The work demonstrates good originality and practical relevance.

The rebuttal addresses several of the concerns raised in the original review, including clarifying training overhead, improving transparency of baseline evaluation, and providing additional controlled comparisons, strengthening confidence in the experimental setup and reported gains.

Some limitations remain, particularly regarding broader analysis of convergence behavior and practical settings such as warm-start inference. These relate to scope and depth rather than correctness.

Overall, the rebuttal reinforces my prior assessment. I maintain my score.

**Key Questions For Authors:**

1. At NFE=1 on ImageNet, DEQ, HyperDEQ, and MDEQ achieve near-random Top-1 accuracies (1.17%, 5.46%, 1.44%), while C-DEQ reports 47.12%. Were these baselines evaluated from standard equilibrium-trained checkpoints, or were they adapted for truncated inference? The training and evaluation protocol for all models should be explicitly stated.

2. Appendix A.2 states the ImageNet model is trained with "sequence length 60" and "validation sequence length 150." What do these terms refer to in an image classification setting?

3. Figure 2 shows near-instantaneous convergence on graph tasks but notably slower refinement on NLP and ImageNet. The paper attributes this to backbone well-posedness without further analysis. Does this difference in convergence behavior affect how the consistency distillation framework should be interpreted across domains?

**Limitations:**

The paper does not adequately discuss its methodological limitations. The approach relies heavily on Anderson Acceleration for both trajectory generation and as a structural prior in the student, and its behavior under alternative DEQ solvers is not evaluated. Fixing (z_0 = 0) yields a deterministic trajectory but may not reflect warm-start inference settings commonly used in practice. The cost of trajectory generation and distillation training is not quantified, and the paper offers limited theoretical analysis of approximation error or convergence under repeated application of the learned map.

**Strengths And Weaknesses:**

**Strengths**

The paper resolves the ambiguity of applying consistency distillation to DEQs by fixing both the initialization and solver (Anderson Acceleration, AA), thereby defining a deterministic FP-ODE trajectory for training. Beyond using AA as a teacher solver, its update structure is incorporated into the student parameterization; the ablation in Fig. 3 indicates that removing this structure leads to significantly slower convergence. Table 4 shows that neither the global nor local consistency objective alone achieves optimal performance, while their combination (λ₁ = 0.8) yields the lowest perplexity. The evaluation reports NFE, wall-clock latency, and GPU memory, demonstrating meaningful inference-time reductions relative to HyperDEQ with relatively stable memory usage across steps. The method is evaluated on NLP, vision, and graph benchmarks, with consistent improvements observed across domains.

**Weakness**

1. The paper does not report trajectory generation or distillation training costs. Appendix B.2 indicates that trajectories are cached but provides no quantitative information on time, storage, or training overhead.

2. At NFE = 1 on ImageNet, the reported Top-1 accuracies for DEQ, HyperDEQ, and MDEQ are very low (1.17%, 5.46%, and 1.44%, respectively), while C-DEQ achieves 47.12%. The paper offers no explanation for this gap, and it is unclear whether baselines were configured for truncated inference or trained under equivalent conditions.

3. While Section 4.3 ablates the loss coefficients λ₁ and λ₂, several other hyperparameters central to the method have no ablations or stated default values: γ (Eq. 5), ρ (Eq. 6), β (Appendix B.3), and p_aug, k_min, k_tail (Algorithm 4). Without guidance on these choices, reproducing the results is non-trivial.

4. Section 3.1 fixes (z_0 = 0) to define a deterministic trajectory, while Algorithm 4 samples (z_0 \sim N(0,I)). The relationship between these two initialization choices is never explicitly clarified in the paper.

5. Appendix A.2 describes the ImageNet model as trained with "sequence length 60" and "validation sequence length 150" — terminology associated with the language modeling setup that is unexplained in an image classification context.

6. AA is used both to generate teacher trajectories and as a structural prior in the student. Whether the approach remains effective with other solvers such as Broyden or standard fixed-point iteration is not evaluated.

---

> ### Author Rebuttal · Authors · 2026-03-31
>
> We thank the reviewer for your constructive comments and address your concerns below.
>
> **Training Overhead:** We provide comprehensive measurements of training overhead across all datasets.
> |Dataset|Storage Mem. (MB/traj.)|Generation Time (s/traj.)|#Point/traj.|Training Time (s/batch)|Peak GPU Mem. (GB)|
> |:-|:-:|:-:|:-:|:-:|:-:|
> |**WikiText-103**| 337.2 | 1.44 | 40 | 3.27 | 17.02 |
> |**ImageNet**| 757.1 | 2.61 | 50 | 4.93 | 20.94 |
> |**ogbn-arxiv**| 173.1 | 0.97 | 30 | 1.21 | 12.90 |
> |**ogbn-products**| 313.9 | 1.34 | 40 | 1.80 | 15.49 |
>
> Generating a 50-point trajectory takes 2.61s/757.1MB on ImageNet (4.93s/20.94GB per training batch), and is even lower on ogbn-arxiv. Overall, the trajectory generation or distillation training costs are remarkably minimal. We will include this full cost analysis in the revised manuscript.
>
> **Clarification on Baseline Configurations at NFE=1:** We *do not* retrain or specially tune baseline models (DEQ, HyperDEQ, MDEQ) for truncated inference. Instead, we use their *official released* weights and code under the same evaluation setting, and directly report their results when constrained to NFE=1 (e.g., 1.17%-5.46% on ImageNet).
>
> This highlights the advantage of our approach: standard DEQ baselines collapse under extreme single-step constraints, whereas C-DEQ learns to directly map inputs toward the equilibrium via consistency distillation. We will add a clarifying note in the experimental section of the revised manuscript.
>
> **Hyperparameter Documentation and Sensitivity Analysis:** We provide default hyperparameter settings and additional ablation studies for all key parameters. Specifically, we set $\gamma=2$ and $\rho=0.1$ for trajectory sampling, and $\beta=0.9$ for multi-step inference. In Algorithm 4, we follow the standard CLLM setup with $p_{aug}=0.1$, $k_{min}=3$, and $k_{tail}=2$. To assess sensitivity, we conduct experiments on WikiText-103 varying $\gamma$, $\rho$, as well as $\beta$.
>
> |PPL↓|$\gamma=0.5$|$\gamma=1.0$|$\gamma=2.0$|$\gamma=5.0$|
> |:-|:-|:-|:-|:-|
> |$\rho=0.05$|93.57|35.29|27.69|51.52|
> |$\rho=0.1$|89.08|34.24|26.43|50.18|
> |$\rho=0.2$|97.23|36.03|30.45|53.29|
>
> |$\beta$|$0.5$|$0.7$|$0.9$|$0.95$|
> |:-|:-|:-|:-|:-|
> |PPL|29.28|28.19|26.43|26.72|
>
> Empirical results show that within a reasonable hyperparameter range, C-DEQ consistently maintains its performance advantage without requiring extensive tuning. A summary table with all defaults and sensitivity results will be added.
>
> **Clarifications on Typos:** We thank the reviewer for carefully identifying these notational and terminology issues.
> - Initialization discrepancy: The initialization in Algorithm 4 is a typographical error. In our framework, the initial state is always set to zero ($z_0 = 0$), ensuring deterministic and reproducible trajectories for consistency distillation. This is consistent with our implementation and provided code.
> - Terminology inconsistency in Appendix A.2: This was an accidental formatting error. The "sequence length 60" and "validation sequence length 150" refer to the WikiText-103 language modeling setting, while for ImageNet the correct input resolution is 224×224 for both training and inference.
>
> We will correct these issues in the revised manuscript and carefully proofread the paper to avoid similar inconsistencies.
>
> **Evaluation of Solver:** Following your suggestion, we clarify that alternative solvers such as Picard and Broyden are technically viable, but empirically exhibit weaker convergence than AA. We conduct a comprehensive comparison against AA across all tasks.
>
> |Method|WikiText-103 (PPL↓ / s/batch)|ImageNet (Acc.↑ / s/batch)|ogbn-arxiv (Acc.↑ / s/batch)|ogbn-products (Acc.↑ / s/batch)|
> |:-|:-:|:-:|:-:|:-:|
> |AA|26.43 / 0.37|74.04 / 0.87|71.40 / 0.16|76.55 / 3.85|
> |Broyden|26.97 / 0.43|71.82 / 1.05|67.83 / 0.17|74.25 / 4.23|
> |Picard|28.68 / 0.33|69.46 / 0.85|61.13 / 0.14|63.91 / 3.56|
>
> The results show AA provides the optimal performance-efficiency trade-off, confirming it as C-DEQ's effective trajectory generator.
>
> **Impact of Teacher Well-Posedness on Convergence:** We appreciate this insightful observation regarding the convergence behavior in Fig. 2. We clarify that the differences are not caused by data domain, but by the intrinsic properties of the teacher DEQ models. Since C-DEQ is initialized from the teacher DEQ and trained via trajectory distillation, it inherently inherits structural properties of the teacher, including its degree of well-posedness. In graph tasks, the teacher IGNN is explicitly designed to ensure strong well-posedness, which leads to the observed near-instant convergence. In contrast, teacher DEQs used in NLP and Vision do not always enforce such strict properties during pretraining, resulting in slower convergence behaviors.
>
> Thus, the observed variation reflects differences in teacher model quality rather than domain-specific effects. We will clarify this point and add discussion in the revised manuscript.

---

> > ### Author Rebuttal · Reviewer_U1DK · 2026-04-03
> >
> > Thank you for the detailed rebuttal. Most concerns are satisfactorily addressed. The added solver comparison partially addresses the concern, but it would help to explicitly clarify whether all solvers were evaluated under an identical experimental setup, including matched NFE settings, to ensure a fully controlled comparison.
> >
> > I maintain my score.

---

> > > ### Author Response · Authors · 2026-04-03
> > >
> > > We appreciate the reviewer's feedback and the opportunity to provide further clarification.
> > >
> > > We would like to clarify that all solvers are evaluated under strictly identical experimental conditions and parameter settings. This includes the NFE, hyperparameters (e.g., $\gamma_1, \gamma_2$), and the underlying computational environment like learning rates. Crucially, we strictly match the NFE settings across all solvers for each dataset to ensure a fully controlled and fair comparison.
> > >
> > > To further address your concern, we have summarized the performance and efficiency under identical settings across the four datasets in the tables below:
> > >
> > >
> > >
> > > **Table 1. Comparison on WikiText-103 with identical settings: $\gamma_1=0.8$, $\gamma_2=0.05$, and training settings (LR=3e-4, WD=0, optimizer=Adam)**
> > >
> > > | NFE | AA (Ours) (PPL $\downarrow$) | AA (Ours) (s/batch $\downarrow$) | Broyden (PPL $\downarrow$) | Broyden (s/batch $\downarrow$) | Picard (PPL $\downarrow$) | Picard (s/batch $\downarrow$) |
> > > | :--- | :---: | :---: | :---: | :---: | :---: | :---: |
> > > | **NFE = 1** | 47.90 | 0.05 | 55.83 | 0.07 | 68.59 | 0.04 |
> > > | **NFE = 2** | 38.98 | 0.09 | 44.76 | 0.15 | 50.02 | 0.08 |
> > > | **NFE = 8** | 26.43 | 0.37 | 26.97 | 0.43 | 28.68 | 0.33 |
> > >
> > > ---
> > >
> > > **Table 2. Comparison on ImageNet with identical settings: $\gamma_1=0.6$, $\gamma_2=0.1$, and training settings (LR=0.05, WD=5e-5, optimizer=SGD)**
> > >
> > > | NFE | AA (Ours) (Top-1 Acc. $\uparrow$) | AA (Ours) (s/batch $\downarrow$) | Broyden (Top-1 Acc. $\uparrow$) | Broyden (s/batch $\downarrow$) | Picard (Top-1 Acc. $\uparrow$) | Picard (s/batch $\downarrow$) |
> > > | :--- | :---: | :---: | :---: | :---: | :---: | :---: |
> > > | **NFE = 1** | 47.12 | 0.52 | 40.97 | 0.65 | 39.76 | 0.51 |
> > > | **NFE = 2** | 58.28 | 0.69 | 51.08 | 0.81 | 49.70 | 0.66 |
> > > | **NFE = 8** | 74.04 | 0.87 | 71.82 | 1.05 | 69.46 | 0.85 |
> > >
> > > ---
> > >
> > > **Table 3. Comparison on Ogbn-Arxiv with identical settings: $\gamma_1=0.8$, $\gamma_2=0.05$, and training settings (LR=0.01, WD=5e-4, optimizer=Adam)**
> > >
> > > | NFE | AA (Ours) (Acc. $\uparrow$) | AA (Ours) (s/batch $\downarrow$) | Broyden (Acc. $\uparrow$) | Broyden (s/batch $\downarrow$) | Picard (Acc. $\uparrow$) | Picard (s/batch $\downarrow$) |
> > > | :--- | :---: | :---: | :---: | :---: | :---: | :---: |
> > > | **NFE = 1** | 56.81 | 0.05 | 48.71 | 0.06 | 43.37 | 0.05 |
> > > | **NFE = 2** | 67.48 | 0.08 | 51.59 | 0.10 | 48.40 | 0.07 |
> > > | **NFE = 5** | 71.40 | 0.16 | 67.83 | 0.17 | 61.13 | 0.14 |
> > >
> > > ---
> > >
> > > **Table 4. Comparison on Ogbn-Products with identical settings: $\gamma_1=0.8$, $\gamma_2=0.1$, and training settings (LR=0.01, WD=5e-4, optimizer=Adam)**
> > >
> > > | NFE | AA (Ours) (Acc. $\uparrow$) | AA (Ours) (s/batch $\downarrow$) | Broyden (Acc. $\uparrow$) | Broyden (s/batch $\downarrow$) | Picard (Acc. $\uparrow$) | Picard (s/batch $\downarrow$) |
> > > | :--- | :---: | :---: | :---: | :---: | :---: | :---: |
> > > | **NFE = 1** | 60.15 | 1.39 | 56.38 | 1.46 | 49.64 | 1.30 |
> > > | **NFE = 2** | 69.68 | 1.86 | 59.76 | 1.97 | 54.81 | 1.73 |
> > > | **NFE = 10** | 76.55 | 3.85 | 74.25 | 4.23 | 63.91 | 3.56 |
> > > ---
> > >
> > > The results consistently show that, under identical settings, AA achieves a superior performance-efficiency trade-off within the C-DEQ framework. We hope this clarification  addresses your concern regarding the controlled comparison.

---

### Official Review · Reviewer_b6c7 · 2026-03-12

**Soundness:** 3
**Presentation:** 2
**Significance:** 2
**Originality:** 2
**Overall Recommendation:** 4
**Confidence:** 4

**Summary:**

This paper proposes a new framework called C-DEQ, aimed at addressing the high inference latency of Deep Equilibrium Models (DEQs). The article reconstructs the iterative solving process of DEQs as a process evolving along a fixed ordinary differential equation (ODE) trajectory toward the equilibrium point and uses consistency distillation to train the model to directly map intermediate states to the final equilibrium point.

Experiments on several large-scale datasets, including WikiText-103, ImageNet, and OGB graph benchmarks, show that under the same few-step inference budget, C-DEQ achieves 2-20 times higher accuracy compared to traditional implicit DEQ methods. It significantly narrows the latency gap between implicit and explicit models while maintaining the constant memory usage advantage of DEQs.

**Compliance With Llm Reviewing Policy:**

Affirmed.

**Final Justification:**

The authors have successfully addressed my main concerns regarding the choice of AA and training complexity. The additional results provided in the revision validate the proposed method.

**Key Questions For Authors:**

*   Have the authors compared this approach against using trajectories generated by standard Fixed-Point Iteration (FPI) (i.e., $z_{k+1} = f(z_k)$) as the teacher signal? Given that FPI trajectories often converge slower and may exhibit oscillations, what is the quantitative performance drop if C-DEQ is distilled from such trajectories instead of AA trajectories?
*    Is the observed performance gain derived from the specific inductive bias of the AA mathematical formulation, or simply from the availability of additional historical information? Have the authors attempted to replace the AA-structured module with a generic MLP or RNN that takes the same sequence of historical states $[z_t, z_{t-1}, \dots, z_{t-m}]$ as input, allowing the network to learn the combination strategy end-to-end without enforcing the AA update rule?
*   The history window size ($m$) is a core parameter of the AA algorithm. Have the authors evaluated the sensitivity of the final performance to different values of $m$ (e.g., $m=2, 4, 6, 8$)? Is there a "saturation point" beyond which increasing the history length yields diminishing returns or leads to overfitting? Furthermore, does the optimal $m$ vary significantly across different domains (e.g., NLP vs. Vision vs. Graphs)?
*  How were the density parameter $\rho$ and the boundary coefficient $\gamma$ in the time mapping function determined? The paper mentions an exponential mapping ($1 - e^{-\rho k}$); were linear or other non-linear mappings considered? Given that $\rho$ heavily weights the early iterations, is there a sensitivity analysis confirming that the current choice is globally optimal rather than a local optimum?
*   To facilitate reproduction of the baseline results, the authors state they "strictly maintained the configurations specified in their respective original implementations." Given the significant variations in default settings across different papers (e.g., learning rate schedulers, warmup strategies, weight decay), could the authors provide a unified hyperparameter table in the appendix listing the exact settings used for all baselines and C-DEQ within *this* experimental environment? This would greatly assist the community in verifying the fairness of the comparison.

**Limitations:**

Computational Overhead of AA: While C-DEQ aims to reduce inference steps, the Anderson Acceleration process itself requires storing historical states and performing linear algebra operations (e.g., solving least-squares problems) at each step during training and potentially during specialized inference. The authors should explicitly discuss the memory footprint and computational cost of maintaining the history window $m$ compared to standard DEQ or fixed-depth models, especially for very large models or long sequences.

**Strengths And Weaknesses:**

**Strengths:**

- The authors connect the discrete iterative solving process of DEQ with the continuous-time ODE trajectory, skillfully addressing the issue of unclear consistent distillation objectives caused by DEQ's path independence. This theoretical reconstruction provides a new paradigm for accelerating implicit models.
- The experimental results are highly convincing. C-DEQ achieves, and in some cases surpasses, the performance of traditional DEQ that requires dozens of iterations, with only 1-8 function evaluations (NFE). Particularly in graph tasks, the convergence is extremely fast, reaching near-equilibrium almost instantly.
- The paper validates the method not only in the NLP domain but also successfully extends it to computer vision and graph neural network domains, demonstrating the framework's generality and scalability.

**Weakness**

- Training Overhead and Complexity: Although inference speed is significantly improved, the training process seems to become more complex. It requires pre-training a teacher DEQ model, generating and caching a large amount of trajectory data, and then performing distillation. Additionally, the introduction of EMA (Exponential Moving Average) and complex loss balancing ($\lambda_1, \lambda_2$) increases the difficulty of hyperparameter tuning. The article does not discuss in detail the proportion of increase in total training time.
- The core premise of the paper relies on reformulating the DEQ solving process as an ODE evolution along a fixed Anderson Acceleration (AA) trajectory. However, AA is just one of many methods for accelerating fixed-point iterations.
- Dependence on Teacher Trajectories: The effectiveness of the method heavily relies on the quality of the teacher trajectories. If the initial DEQ model converges slowly or is unstable, the generated trajectories may not be optimal, thus limiting the potential of the student model. The article mainly focuses on acceleration and discusses little about the performance bottleneck transmission from the teacher model itself.
-  Although C-DEQ performs well in low-step scenarios, in single-step (NFE=1) inference, its performance still lags behind multi-step inference or explicit SOTA models (such as Transformer-XL or Inception-V2) to a certain extent (for example, on ImageNet, the single-step Top-1 accuracy is about 47%, while multi-step can reach 74%). This indicates that to achieve optimal performance, multiple iterations are still required, which may be a compromise in certain extremely low-latency scenarios.

---

> ### Author Rebuttal · Authors · 2026-03-31
>
> We greatly appreciate the constructive comments and address your concerns below. All additional results are provided at https://anonymous.4open.science/r/ICML2026_Rebuttal_table_a-4640/Results.md and will be included in the revised manuscript.
>
> **Training Overhead and Complexity:**
> 1. Pre-training: We clarify that C-DEQ focuses only on the distillation stage and does not include DEQ pre-training. We directly use the weights of a pretrained DEQ as initialization, following standard practice in prior work such as HyperDEQ and Skip-DEQ. We additionally compare the total wall-clock training time under the same setting in *Table A*. The results show that the training time required by C-DEQ accounts for only a small fraction of that of DEQ.
> 2. Training overhead: The overhead from trajectory caching and distillation is minimal. As shown in *Table B*, generating a 50-step ImageNet trajectory takes only 2.61s and 757.1 MB, and the distillation stage adds only modest cost (~10% of teacher training time). Importantly, this small overhead leads to a remarkable reduction in inference latencies.
> 3. EMA and hyperparameter tuning: EMA is adopted directly as a standard technique in CD [3,4]. We introduce it just for reproducibility. The terms in Eq. 11 are complementary and all necessary (see Lines 259-270). Table 4 confirms the importance of each component. This ablation study further demonstrates that $\lambda_{1}$, which balances the global/local consistency, achieves near-optimal performance over a relatively wide range around 0.8. Similarly, $\lambda_{2}$, which controls task-level regularization, only requires a small value (e.g., 0.05) to be effective. These observations indicate that both global/local consistency and task-level regularization can be reliably applied without extensive search.
>
> **Choice of AA:** Alternatives such as Picard and Broyden's methods are also applicable. We choose AA due to its faster convergence and established effectiveness in DEQ literature [1,5]. We additionally conduct a direct comparison in *Table C*. These results show that AA achieves the best performance-efficiency trade-off across tasks, i.e., AA is not an arbitrary choice, but the most effective trajectory generator for preserving the performance of C-DEQ.
>
> **Dependence on Teacher Trajectories:** (1) Slow or unstable convergence is a general challenge of DEQ pre-training rather than a limitation specific to our distillation method, and has been largely mitigated by prior works [6-8] while inference speed remains a key bottleneck which our work targets. (2) C-DEQ reduces sensitivity to imperfect trajectories via AA and augmented trajectory sampling. As shown in Fig. 2a-b, C-DEQ converges in ~10 steps even when the teacher converges slowly.
>
> **Single-Step Inference:** Single-step inference is extremely challenging for DEQs due to the inherent infinite depth, but C-DEQ substantially outperforms previous implicit SOTAs with NFE=1, thereby significantly narrowing the gap to explicit models. Moreover, C-DEQ demonstrates a favorable accuracy-efficiency trade-off for most real-world scenarios, e.g., for NFE = 8, C-DEQ outperforms prior implicit SOTAs while approaching explicit models.
>
> **Ablation on AA:** Following your suggestion, we conduct an ablation replacing the AA module with generic MLP and RNN. The corresponding code and results (*Table D*) are in the link. Both MLP and RNN fail to match the convergence speed and accuracy of the AA-based design.
> These results indicate that simply providing historical information is insufficient; the structured inductive bias of AA is the key factor behind performance gains.
>
> **Sensitivity on $m$:** We conduct an ablation study in *Table E* to evaluate how $m$ affects results. Increasing $m$ yields only slight accuracy gains without clear saturation, but incurs much higher inference latency. A small $m$ provides the best accuracy-efficiency trade-off.
>
> **Hyperparameter:**
> (1) We adopt the exponential mapping in $t_k$ following CM setup, and validate it against linear and cosine mappings. Empirically (in *Table F*), it outperforms linear and cosine mappings. (2) We determine $\rho$ and $\gamma$ via small-scale grid search.  *Table G* shows a convex performance trend, indicating stable behavior and low tuning sensitivity. (3) We provide  hyperparameter configurations used for all baselines and C-DEQ across datasets in *Figure A*.
>
> References:
>
> [1] Bai, S., et al. Neural deq solvers. In *ICLR* 2021.
>
> [2] Pal, A., et al. Continuous deqs. arXiv preprint 2022.
>
> [3] Song, Y., et al. Consistency models. In *ICML* 2023.
>
> [4] Geng, Z., et al. Consistency models made easy. arXiv preprint 2024.
>
> [5] Baker, J., et al. IGNNs: A monotone operator viewpoint. In *ICML* 2023.
>
> [6] Bai, S., et al. Stabilizing equilibrium models by Jacobian regularization. In *ICML* 2021.
>
> [7] Rahmani, B., et al. Regularizing the infinite. In *NeurIPS* 2024.
>
> [8] Geng, Z., et al. On training implicit models. In *NeurIPS* 2021.

---

> > ### Author Rebuttal · Reviewer_b6c7 · 2026-04-03
> >
> > The authors have successfully addressed my main concerns regarding the choice of AA and training complexity. The additional results provided in the revision validate the proposed method. I will raise my score.

---

> > > ### Author Response · Authors · 2026-04-07
> > >
> > > Thank you for your positive final assessment. We deeply appreciate your thorough and insightful feedback. All additional results and discussions will be included in the revision.

---

### Official Review · Reviewer_Gm7Q · 2026-03-13

**Soundness:** 3
**Presentation:** 3
**Significance:** 2
**Originality:** 3
**Overall Recommendation:** 5
**Confidence:** 4

**Summary:**

This paper uses a consistency type distillation of a pretrained deep-equilibrium model (DEQ) to achieve faster inference. This is achieved by making use of the observation that the equilibrium state of the DEQ can be characterized by an ODE which can be fit by a short-cut map. Furthermore, they choose to construct teacher trajectories using the Anderson accelerator dynamics and parametrize the student model accordingly, leading to better equilibrium state convergence. The distilled DEQ is evaluated on several task involving text generation, image classification and graph node classification, showing increased tradeoff of performance and inference speed.

**Compliance With Llm Reviewing Policy:**

Affirmed.

**Final Justification:**

The authors have addressed all of concerns and the gap between the ODE (2) and the Anderson acceleration solver used for consistency training.

**Key Questions For Authors:**

- The explanation for why the authors use the AA solver instead of the ODE is confusing to me. They claim the reason is they need discrete intermediate samples of states, yet any numerical solver of that ODE will give you that. It seems like the AA solver is used simply because it converges more accurately for DEQ models? I think it would make the paper stronger to explain this differently and what it means for the consistency framework to switch from the ODE to AA.
- Is it possibly within scope to increase training efficiency with a consistency model?
- Even though the focus of the experiments comparing DEQ models, some inference times are reported for explicit networks, but most are not. I think it would be helpful to know the wall-clock time for the explicit models (or as many as possible).
- Could the authors include more information about training time of distillation to better understand the additional compute overhead compared to the original pretraining?

**Limitations:**

yes

**Strengths And Weaknesses:**

Strengths
- Using the ODE characterization of the equilibrium state to learn the consistency map is an interesting insight and well-motivated by the goal of faster inference.
- Parameterizing the student model with the anderson accelerator appears to be a novel and well-motivated architecture choice.
- The experimental results are very thorough covering multiple problem domains (text, images, graphs). The results do show that the distilled C-DEQ is able to tradeoff performance and speed better than other DEQ approaches.

Weaknesses

- This method requires a pretrained DEQ to distill for faster inference. On the other hand, training DEQ is still arguably the main bottleneck preventing scaling to larger datasets. Faster inference will not address the performance gap (even at high NFE) to explicit methods which can only be achieved by scaling up training.
- Inference is still slow compared to explicit architectures at acceptable levels of model accuracy (at least for Transformer XL, the only inference time for an explicit method reported).
- The use of the Anderson accelerator dynamics in place of the fixed-point ODE does leave a slight conceptual gap in the theoretical motivation.

---

> ### Author Rebuttal · Authors · 2026-03-31
>
> We thank the reviewer for your careful consideration. We greatly appreciate the constructive comments and address your concerns below.
>
> **Pretrained DEQ Bottleneck Limits Impact:** As stated in Lines 11-15 and 67-73,  we would like to emphasize that our objective is strictly inference acceleration, not scaling up training. This is a challenging task even for existing pretrained DEQs on relatively modest-sized datasets. Our method effectively resolves this bottleneck.
> Moreover, recent literature demonstrates that implicit models have potential; scaling to large datasets. For example, Schöne et al. (2025) demonstrate competitive performance with Implicit LLMs (1.3B). Furthermore, Kou et al. (2024) proposed CLLMs (6.7B), which formulate the next-token prediction as a Jacobi FP problem and achieve competitive performance.  Our acceleration method has immense potential to contribute to future large-scale models. Thank you for bringing this up. It provides valuable guidance for our future work.
>
> **Comparison with Explicit Models:** Our core contribution is drastically mitigating the inherent latency bottleneck in DEQs, which stems from their iterative nature. For example, on WikiText-103, C-DEQ significantly improves the few-step performance of prior work (e.g., HyperDEQ), reducing PPL from 70.19 to 47.90 at NFE=1 and from 51.31 to 38.98 at NFE=2. At NFE=8, C-DEQ further reduces PPL from 31.37 to 26.43, achieving performance comparable to Transformer-XL. Although it remains slower than Transformer-XL (0.13s), C-DEQ *substantially narrows the latency gap*, reducing inference time to 0.37s compared to 1.21s for HyperDEQ. These results demonstrate that C-DEQ effectively bridges the efficiency gap between prior DEQ methods and explicit models.
>
> **The motivation of AA:**  As stated in Lines 125-128, CD requires discrete intermediate states while FP-ODE defines a continuous path, and AA provides an effective discretization that significantly accelerates convergence (Lines 130-134). Furthermore, while alternative solvers such as Picard and Broyden's methods are viable, we empirically find their convergence to be noticeably inferior to AA, as shown in the table below.
>
> |Method|WikiText-103 (PPL↓ / s/batch)|ImageNet (Acc.↑ / s/batch)|ogbn-arxiv (Acc.↑ / s/batch)|ogbn-products (Acc.↑ / s/batch)|
> |:-|:-:|:-:|:-:|:-:|
> |AA|26.43 / 0.37|74.04 / 0.87|71.40 / 0.16|76.55 / 3.85|
> |Broyden|26.97 / 0.43|71.82 / 1.05|67.83 / 0.17|74.25 / 4.23|
> |Picard|28.68 / 0.33|69.46 / 0.85|61.13 / 0.14|63.91 / 3.56|
>
> As shown in the table, AA provides the optimal trade-off between performance and efficiency.  We will add these ablation studies  to the revised manuscript.
>
> **Potential for Increasing Training Efficiency:** Increasing training efficiency for C-DEQ is  promising. Our C-DEQ distills a pretrained DEQ by fine-tuning,  thus it is  compatible with  efficient training methods, e.g., LoRA. Furthermore, advanced CD strategies, e.g., the noise schedules and loss weighting schemes introduced in ECT can be applied. Exploring these methods represents a valuable direction for our future work. Thank you for your valuable questions.
>
> **Wall-Clock Times for Explicit Baselines:** Following your suggestion, we have comprehensively tested the inference latency of explicit models across all modalities under the identical environment used for C-DEQ. We have updated Tables 1-3 (see https://anonymous.4open.science/r/ICML2026_Rebuttal_tables-4640/Tables1-3.md). These additional results support our conclusion: C-DEQ achieves competitive performance with low inference latency.
>
> **Quantitative Analysis of Distillation Training Overhead:** Following your suggestion, we comprehensively benchmarked the costs of trajectory generation and distillation training in the following table.
> |Dataset|Storage Mem. (MB/traj.)|Generation Time (s/traj.)|#Point/traj.|Training Time (s/batch)|Peak GPU Mem. (GB)|
> |:-|:-:|:-:|:-:|:-:|:-:|
> |**WikiText-103**| 337.2 | 1.44 | 40 | 3.27 | 17.02 |
> |**ImageNet**| 757.1 | 2.61 | 50 | 4.93 | 20.94 |
> |**ogbn-arxiv**| 173.1 | 0.97 | 30 | 1.21 | 12.90 |
> |**ogbn-products**| 313.9 | 1.34 | 40 | 1.80 | 15.49 |
>
> These results indicate that C-DEQ demonstrates high training efficiency. The teacher trajectory generation is very efficient and acts as a strict one-time overhead since trajectories are cached offline. During distillation, peak GPU memory is  manageable. The distillation training time (4.93 s/batch for ImageNet, and 3.27 s/batch for WikiText-103) requires only ~10% of the computational time of the original teacher DEQ training (29.37 s/batch for ImageNet). We will include these results in the revised manuscript.
>
> **References:**
> - Schöne M., et al. Implicit language models are RNNs. In *ICML* 2025.
> - Kou, S., et al. CLLMs: Consistency large language models. In *ICML* 2024.
> - Geng, Z., et al. Consistency models made easy. In *ICLR* 2025.

---

> > ### Author Rebuttal · Reviewer_Gm7Q · 2026-04-03
> >
> > Thank you for the detailed response. I think its understandable that consistency-based pretraining is not the scope of this work and would hope to see your method eventually applied to these new large-scale DEQ approaches for LLMs.
> >
> > Regarding the AA solver, I believe I understand the cited text already and I don't really have doubts about the empirical performance of the AA solver over other discretizations. The consistency approach is motivated by the FP-ODE (2) and requires a discretized teacher trajectory. One discretizer is a simple Euler-integration of the FP-ODE, which would correspond to that ODE in the continuous-time limit. Even some of the proofs of the consistency models paper [Song2023] rely on the properties of this limiting ODE. However, its not obvious to me that this is true for the AA solver which directly addresses eq (1) and is only related to the FP-ODE (2) by sharing the same fixed point.
> >
> > I don't believe this "breaks" the method in any way (you can probably still define the consistency approach at a trajectory solution level as long as the AA solver is well-behaved enough), but I think leaving this conceptual gap in the introduction of the method is really confusing. I would even consider trying to reformulate the consistency method directly from the fixed-point equation (1), rather than posing this ODE concept which is not really used. At very least, I would strongly recommend trying to explain the gap in this analogy more clearly in section 3.
> >
> > I appreciate your further discussion and clarification.
> >
> > [Song2023] Song et al Consistency Models, ICML 2023.

---

> > > ### Author Response · Authors · 2026-04-06
> > >
> > > We greatly appreciate your deep understanding! We would like to clarify that our ODE perspective does not rely on a specific ODE form, but rather on the existence of a well-defined (path-dependent and continuous) trajectory to motivate the consistency framework. From your constructive feedback, we realize that the mismatch between the specific formulation of Eq. (2) and the AA solver will cause confusion. This is because different solvers induce different continuous-time dynamics in the limit while they all share the same fixed point. For example, a simple Picard iteration corresponds to a first-order FP-ODE, while AA can be interpreted as inducing a history-dependent continuous-time dynamics [1].
> > >
> > >
> > > To address this conceptual gap and to avoid confusion, we will make the following *minor* modifications:
> > > 1. Clarify that Eq. 2 serves only as a **conceptual illustration**, and replace it with a  general formulation:
> > > $$\frac{d\mathbf{z}(t)}{dt} = G(\mathbf{z}_t, t; \boldsymbol{\theta})$$
> > > where $G$ is only a conceptual function that represents the solver-induced dynamics (which may be history-dependent).
> > >
> > > 2. Provide a table illustrating the correspondence between different solvers (e.g., Picard, Newton, AA) and their associated continuous-time dynamics.
> > >
> > >
> > > | Solver         | Discrete Update | Corresponding FP-ODE (Continuous-Time) |
> > > |:----------------|:----------------:|:---------------------------------------:|
> > > | **Picard**     |    $$z_{k+1} = f(z_k, x)$$     |     $$\dot{z}(t) = f(z, x) - z$$              |
> > > | **Quasi-Newton** |      $$z_{k+1} = z_k - B_k^{-1} (f(z_k, x) - z_k), B_k \approx J_f - I$$    |        $$\dot{z}(t) = - B(t)^{-1} (f(z, x) - z), \ B(t) \approx J_f - I$$            |
> > > | **AA (m=1)**   |$$z_{k+1} = \beta \sum_{i=0}^{m} \alpha_i f(z_{k-m+i}, x) + (1-\beta) \sum_{i=0}^{m} \alpha_i z_{k-m+i}, \quad \sum_{i=0}^m \alpha_i = 1$$    |           $$\dot{z}(t) = \frac{\beta}{1 + \alpha(t)(1-\beta)} [ e(t) - \alpha(t)\dot{f}(z,x) ],$$ $$ \alpha(t) = \frac{\dot{e}(t)^\top e(t)}{\|\dot{e}(t)\|^2}, \ e(t) = f(z,x) - z$$        |
> > >
> > >
> > >
> > > > Note: AA operates via dynamic residual projection from history, where $\alpha_i$ represents the dynamic mixing weights and the damping factor $\beta \in (0, 1)$ stabilizes non-linear overshooting, which is discussed in detail in Appendix B.1 of the manuscript. For $m >1$, the underlying logic remains fundamentally consistent with $m =1$. Due to its complexity, we refer it to a recent work [1].
> > >
> > > This modification is conceptual and does not affect the method or experiments. It is intended only to clarify the relationship between solver dynamics and the FP-ODE perspective, and to remove potential confusion.
> > >
> > > We sincerely appreciate your deep understanding of the underlying principles and your constructive feedback, which helped us better present our work more clearly.
> > >
> > > [1] Chen, K., Jiang, Y., & Vuik, K. (2025). The Dynamical Anatomy of Anderson Acceleration: From Adaptive Momentum to Variable-Mass ODEs. arXiv preprint arXiv:2512.21269.

---

### Decision · Program_Chairs · 2026-04-30

**Decision:**

Accept (regular)

**Comment:**

This paper proposes C-DEQ, a consistency training framework for deep equilibrium models (DEQs). The key idea is to interpret the equilibrium finding process as a discrete sampling from a fixed trajectory ODE, and to adapt consistency distillation to learn a direct mapping from intermediate states to the equilibrium. In addition, the method leverages Anderson acceleration (AA) both for generating teacher trajectories and as a structured parameterization for the student model.

Most reviewers find the technical contribution to be novel and the overall approach well-motivated. While there are some concerns regarding empirical aspects, the results are generally convincing across multiple domains. There are also concerns about training cost, reliance on pretrained DEQs, and the theoretical conditions for convergence; some of these are only partially addressed during the rebuttal. Nevertheless, I believe that the overall contribution outweighs these remaining concerns.